# Unconventional bipartite entanglement in the quantum dimer magnet $Yb_2Be_2SiO_7$

A. Brassington[1], Q. Ma[2], G. Duan [3], S. Calder [2], A. I. Kolesnikov [2], K. M. Taddei[2,11], G. Sala[4,12], E. S. Choi[5], H. Wang [6], W. Xie [6], B. A. Frandsen[7], N. Li[8], X. F. Sun [8], C. Liu [9,13], R. Yu [3,10], H. D. Zhou [1] ✉ & A. A. Aczel [2] ✉

The quantum dimer magnet, with antiferromagnetic intradimer and interdimer Heisenberg exchange between spin-1/2 moments, is known to host an $(|\uparrow\downarrow\rangle - |\downarrow\uparrow\rangle)/\sqrt{2}$ singlet ground state when the intradimer exchange is dominant. Rare-earth-based quantum dimer systems with strong spin-orbit coupling offer the opportunity for tuning their magnetic properties by using magnetic anisotropy as a control knob. Here, we present bulk characterization and neutron scattering measurements of the quantum dimer magnet $Yb_2Be_2SiO_7$. We find that the $Yb^{3+}$ ions can be described by an effective spin-1/2 model at low temperatures and the system does not show signs of magnetic order down to 50 mK. The magnetization, heat capacity, and neutron spectroscopy data can be well-described by an isolated dimer model with highly anisotropic exchange that stabilizes a singlet ground state with a wavefunction $(|\uparrow\uparrow\rangle - |\downarrow\downarrow\rangle)/\sqrt{2}$ or $(|\uparrow\uparrow\rangle + |\downarrow\downarrow\rangle)/\sqrt{2}$. Our results show that strong spin-orbit coupling can induce unusual entangled states of matter in quantum dimer magnets.

Insulating magnets devoid of ground states with long-range order and instead characterized by strongly-interacting units featuring quantum entanglement[1,2] have attracted intense interest in the field of quantum magnetism. The simplest case consists of a series of dimers with two entangled spins per unit[3,4], which is known as a quantum dimer magnet for spin-1/2. The Shastry-Sutherland lattice (SSL) consists of a plane of orthogonal dimers[5,6], and it is now known as one of the canonical two-dimensional models that can host entangled spin states. For isotropic Heisenberg spins on the SSL with a sufficiently small ratio of interdimer-to-intradimer exchange interactions $\alpha = J_2/J_1$, the Hamiltonian is exactly solvable and the ground state is predicted to be a direct product of entangled spin-singlet states. The gapped elementary excitations associated with this exotic state are known as triplons[7,8].

SSL research progressed beyond the initial theoretical studies of the simple toy model with the discovery of $SrCu_2(BO_3)_2$[9,10], which consists of weakly-coupled planes of $S = \frac{1}{2}$ $Cu^{2+}$ ions with the desired orthogonal dimer geometry[11]. The $\alpha$ ratio is sufficiently small to generate a spin-singlet ground state[9,10], although it is close to the critical value required to produce Neel antiferromagnetic order[11,12]. This

[1]Department of Physics and Astronomy, University of Tennessee, Knoxville, TN, USA. [2]Neutron Scattering Division, Oak Ridge National Laboratory, Oak Ridge, TN, USA. [3]School of Physics and Beijing Key Laboratory of Optoelectronic Functional Materials and Micro-nano Devices, Renmin University of China, Beijing, PR China. [4]Oak Ridge National Laboratory, Oak Ridge, TN, USA. [5]National High Magnetic Field Laboratory and Department of Physics, Florida State University, Tallahassee, FL, USA. [6]Department of Chemistry, Michigan State University, East Lansing, MI, USA. [7]Department of Physics and Astronomy, Brigham Young University, Provo, UT, USA. [8]Anhui Provincial Key Laboratory of Magnetic Functional Materials and Devices, Institutes of Physical Science and Information Technology, Anhui University, Hefei, Anhui, PR China. [9]School of Engineering, Dali University, Dali, Yunnan, PR China. [10]Key Laboratory of Quantum State Construction and Manipulation (Ministry of Education), Renmin University of China, Beijing, PR China. [11]Present address: X-ray Sciences Division, Advanced Photon Source, Argonne National Laboratory, Lemont, Illinois, USA. [12]Present address: J-PARC Center, Japan Atomic Energy Agency (JAEA), Materials and Life Science, Ibaraki, Japan. [13]Present address: School of Physics and Mechatronic Engineering, Guizhou Minzu University, Guiyang, Guizhou, PR China. ✉e-mail: hzhou10@utk.edu; aczelaa@ornl.gov

proximity to a quantum phase transition has now been investigated in great detail by both experiment and theory, with a pressure-induced singlet plaquette phase first identified by neutron spectroscopy[13] and multiple theoretical studies predicting quantum spin liquid phases in this regime as well[14–17]. $SrCu_2(BO_3)_2$ also exhibits striking magnetic-field-induced phenomena, with several magnetization plateaus observed that were first explained by triplon crystallization[18–20]. The lower-field plateaus were later argued to arise from triplon bound state crystallization instead[21].

More recently, the effect of strong spin-orbit-coupling on the magnetic properties of the SSL has received significant attention. Relevant theoretical work has explored the magnetic properties of the Ising or XXZ SSL Hamiltonian[22–27], which can be realized for systems with significant magnetic anisotropy. On the experimental side, the $RB_4$ ($R$ = rare earth) family members were initially explored as possible anisotropic SSL systems[28–33]. Unfortunately, they are itinerant magnets with long-range Ruderman–Kittel–Kasuya–Yosida (RKKY) exchange interactions mediated by the conduction electrons[34–38], so their magnetic Hamiltonians are more complex than the simple $J_1$-$J_2$ (or nearest-neighbor and next-nearest-neighbor) SSL model. Two insulating families $BaR_2MX_5$ ($M$ = Zn, Pd, or Pt; $X$ = O or S)[39–46] and $R_2Be_2ZO_7$ ($Z$ = Ge or Si)[47,48] have now come to the forefront as superior model systems for exploring SSL physics in strongly anisotropic magnets. A subset of these rare-earth-based materials host effective spin-1/2 degrees of freedom at sufficiently low temperatures, which arise from a well-isolated crystal field ground state doublet. While ordered ground states have often been identified[40,41,43,45,49–51], effective spin-1/2 systems can realize more exotic behavior. Both $BaCe_2ZnS_5$[46] and $Yb_2Be_2GeO_7$[52,53] are effective spin-1/2 dimer systems with entangled ground states and the latter has emerged as a quantum spin liquid candidate[52].

In this work, we report comprehensive bulk characterization and neutron scattering measurements of the quantum dimer magnet $Yb_2Be_2SiO_7$ with SSL geometry. We find that the $Yb^{3+}$ ions have effective spin-1/2 degrees of freedom at sufficiently low temperatures below 10 K, and we observe no evidence for magnetic order down to dilution-refrigerator temperatures of 50 mK. The strong magnetic anisotropy of this system generates nearly Ising moments with a strong tendency to align along the [001]-axis. The zero-field neutron spectroscopy and the magnetic-field and temperature-dependence of both the magnetization and specific heat data closely match expectations for a novel entangled dimer state. More specifically, $Yb_2Be_2SiO_7$ consists of a series of dimers with $S_z \neq 0$ ground state wavefunctions ($(|\uparrow\uparrow\rangle - |\downarrow\downarrow\rangle)/\sqrt{2}$ or $(|\uparrow\uparrow\rangle + |\downarrow\downarrow\rangle)/\sqrt{2}$. The anisotropic intradimer exchange induced by the strong spin-orbit coupling in this system is responsible for stabilizing this exotic state that cannot be realized by the Heisenberg SSL model for spin-1/2 systems.

## Results
### Crystal structure magnetic susceptibility, and neutron powder diffraction
$Yb_2Be_2SiO_7$ crystallizes into the tetragonal space group $P$-$42_1m$ (113) with room temperature lattice constants $a$ = 7.207(1) Å and $c$ = 4.719(1) Å determined from single crystal x-ray diffraction (see Supplementary Discussion Section I and Supplementary Tables S1, S2 for more details). The eight-fold coordinated $Yb^{3+}$ ions form an SSL lattice in the $ab$-plane with intradimer and interdimer distances of 3.239(1) Å and 3.836(1) Å, respectively, as shown in Fig. 1a. Adjacent $Yb^{3+}$ planes are separated by 4.719(1) Å. A view of the crystal structure along the $c$-axis with all the atoms shown is depicted in Fig. 1b.

The inverse magnetic susceptibility of single crystalline $Yb_2Be_2SiO_7$, plotted as field/magnetization $H/M$, is presented in Fig. 1c between 0.4 K and 300 K. The data was collected in a small magnetic field of 0.1 T applied along the three high-symmetry directions [001], [110], and [100]. The same data is shown in the low-$T$ range between

0.4 K and 1.6 K in Fig. 1d, with the real component of the AC susceptibility data from 0.05 K to 0.65 K also included for two field orientations. There are no clear signatures of long-range magnetic order in this data. Curie–Weiss fits were performed over both high-$T$ (150–300 K) and low-$T$ (1–5 K) ranges. The effective moment values from the high-$T$ fits are broadly consistent with expectations for free $Yb^{3+}$ ions. The low-$T$ fits return effective moments and Curie–Weiss temperatures of 4.22(5) $\mu_B$ and −0.10(2) K for $\mathbf{H}\|[001]$, 1.71(1) $\mu_B$ and −1.32(1) K for $\mathbf{H}\|[110]$, and 1.67(1) $\mu_B$ and −1.21(1) K for $\mathbf{H}\|[100]$. The comparatively large effective moment for $\mathbf{H}\|[001]$ indicates that the moments have a strong preference to align with the crystallographic [001]-axis at low-$T$ and the larger in-plane Curie–Weiss temperatures are consistent with the expected quasi-2D behavior of $Yb_2Be_2SiO_7$. The small negative Curie–Weiss temperatures are indicative of weak anti-ferromagnetic interactions in the system under the assumption of isotropic exchange, which is not valid for this system as shown below.

Unpolarized neutron powder diffraction (NPD) patterns collected at 0.25 K and 2 K are presented in Fig. 1e and f, respectively. Both patterns are well-described by nuclear Bragg peaks associated with the $Yb_2Be_2SiO_7$ crystal structure and the Al sample can. The expected positions of these peaks are indicated by the green and red ticks, respectively. While Rietveld refinements of the data using FULLPROF[54] reveal no evidence for oxygen off-stoichiometry in this sample, they do identify ~4.4% Be/Si site mixing. Most importantly, there are no additional Bragg peaks or enhanced peak intensities that can be attributed to long-range magnetic order in $Yb_2Be_2SiO_7$, even down to 0.25 K. Additional details of the 0.25 K refinement are presented in Supplementary Table S3.

### $Yb^{3+}$ single-ion ground state
Zero-field specific heat data measured with a single crystal and the corresponding magnetic entropy extracted by integrating $C_p/T$ as a function of temperature are shown in Fig. 2a and its inset, respectively. There are no typical features of magnetic ordering in the heat capacity data. Instead, it is dominated by a broad peak centered near 0.5 K, while the magnetic entropy approaches $R\ln(2)$ at 4 K. The entropy result validates the use of an effective spin-1/2 model for the $Yb^{3+}$ ions at low-$T$ and indicates that the broad peak does not have a crystal field origin. Neutron powder spectroscopy, with the scattering intensity plotted as a function of momentum transfer $Q$ and energy transfer $E$ in Fig. 2b, was used to measure the $Yb^{3+}$ crystal field spectrum. The magnetic scattering contribution was isolated by subtracting the scattering from the non-magnetic analog $Lu_2Be_2SiO_7$ measured under the same experimental conditions and scaled appropriately to account for the scattering cross-section difference between Yb and Lu. There is no evidence for low-lying crystal field levels in this data. Instead, there are three excitations centered about 11, 23, and 36 meV as shown in Fig. 2c that likely correspond to the three excited crystal field doublets expected for $J$ = 7/2 Kramers $Yb^{3+}$ ions in a low-symmetry ligand environment. Interestingly, these three excitations are much broader than the instrument energy resolution, which is readily apparent when comparing this data to previous results from $Er_2Be_2SiO_7$[51] with nearly resolution-limited crystal field levels. This broadening may be a consequence of the Be/Si site mixing that we identified in the Rietveld refinements of the HB-2A data. Interestingly, we confirmed that there is no evidence for Be/Si site mixing in the previously reported HB-2A data for $Er_2Be_2SiO_7$[51]. We also collected Pair-Distribution-Function (PDF) data using the time-of-flight powder diffractometer NOMAD to investigate possible local structural distortions in $Yb_2Be_2SiO_7$ (see Supplementary Discussion Section I and Supplementary Fig. S1), but the PDF data is explained well by the global structure.

The $Yb^{3+}$ ions have a monoclinic point group symmetry of $C_s$ with a single mirror plane symmetry element. The crystal field environment of the four $Yb^{3+}$ ions in the chemical unit cell is different, with the $y$-axis of the local coordinate system indicated by the orange arrows in

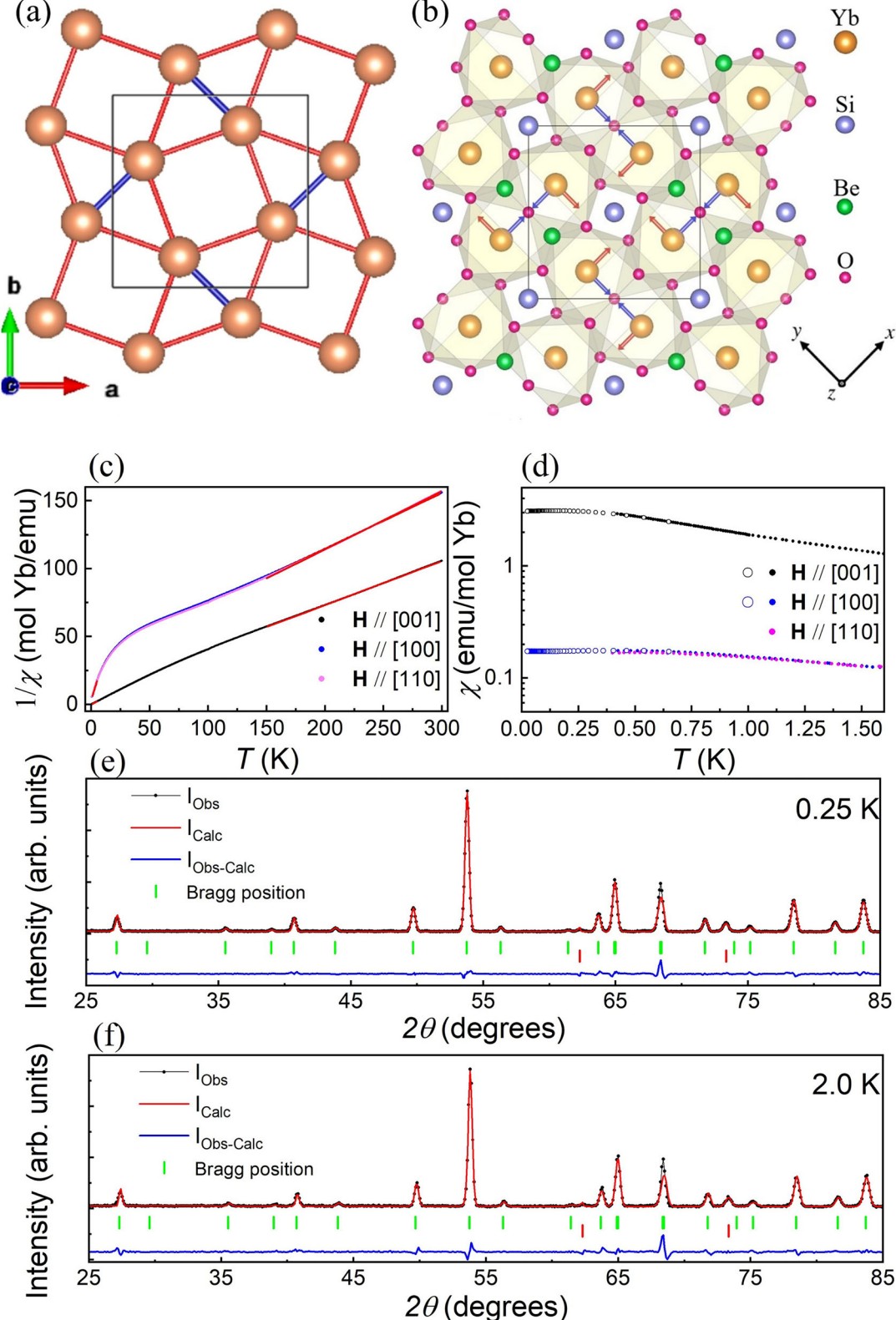

**Fig. 1 | Crystal structure, magnetic susceptibility, and neutron powder diffraction. a** The arrangement of $Yb^{3+}$ ions in $Yb_2Be_2SiO_7$ forming an SSL as viewed along the crystallographic [001]-axis. The intradimer and interdimer bonds $J_1$ and $J_2$ are shown in blue and red, respectively. **b** The full crystal structure of $Yb_2Be_2SiO_7$ viewed along the same axis. The four $Yb^{3+}$ ions in the chemical unit cell have g-tensors with different local axes, as explained in the main text. **c** The inverse susceptibility measured in a 0.1 T field applied along three high-symmetry crystallographic directions. The Curie–Weiss fits in both the high and low-temperature regimes are shown in red. **d** The low-$T$ susceptibility data plotted along three high-symmetry crystallographic directions. Closed circles and open circles represent DC and AC susceptibility data, respectively. **e, f** Neutron powder diffraction patterns collected at 0.25 K and 2 K, respectively. Multi-phase Rietveld refinements with the known $Yb_2Be_2SiO_7$ crystal structure (green ticks) and Al from the sample can (red ticks) explain the data well. No additional peaks associated with long-range order are observed.

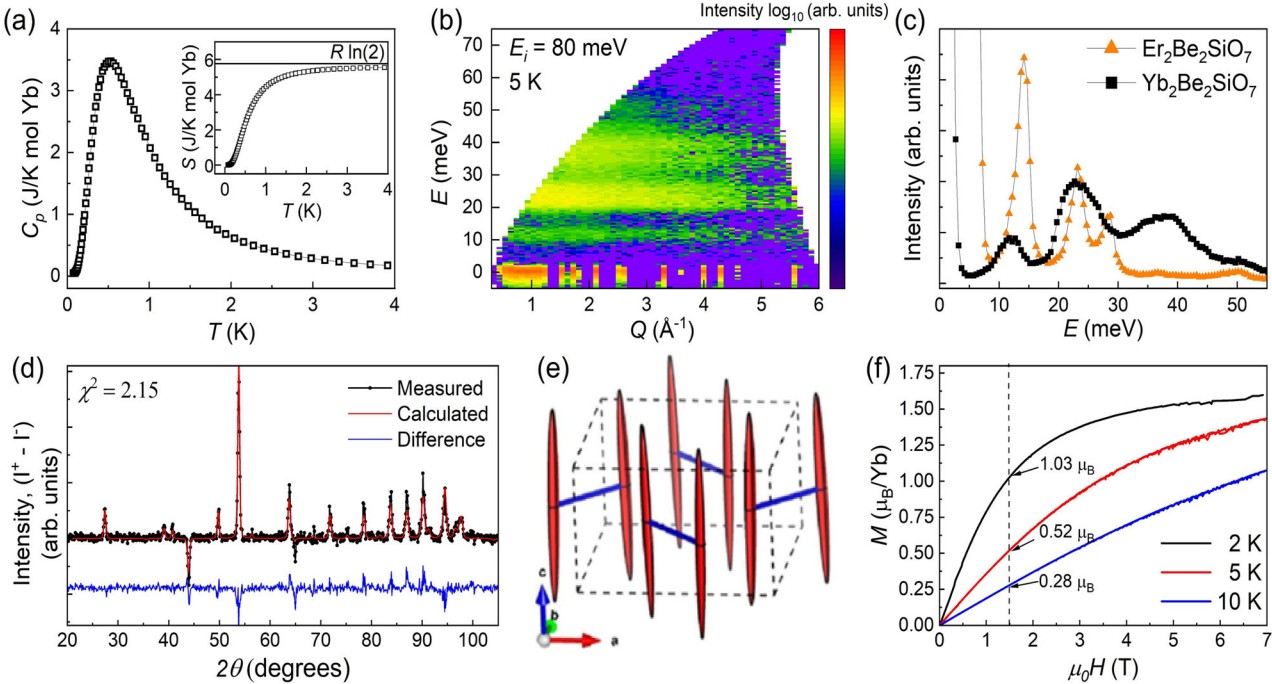

**Fig. 2 | $Yb^{3+}$ single ion ground state. a** Zero-field heat capacity data reveals a broad Schottky anomaly centered near 0.5 K. The inset shows the magnetic entropy over the same temperature range, which approaches $R\ln(2)$ at 4 K. **b** Color contour plot of the scattering intensity for the $Yb^{3+}$ ions in $Yb_2Be_2SiO_7$ as a function of momentum and energy transfer at 5 K from SEQUOIA with $E_i = 80$ meV. This difference map was obtained by subtracting scaled $Lu_2Be_2SiO_7$ data to account for the variation in the Yb and Lu neutron scattering cross sections. **c** Constant-$Q$ cuts ($Q$ integration range [0.25, 5] $Å^{-1}$) of the same $Yb_2Be_2SiO_7$ data (with no Lu subtraction) and the equivalent data for the magnetic $Er^{51}$ analog. **d** The difference pattern, $I^+ - I^-$, obtained from the pNPD measurements at 5 K. The Rietveld refinement is superimposed on the data and the fit residual is shown below it. **e** The magnetization ellipsoids obtained from the pNPD analysis at 5 K. The $Yb^{3+}$ anisotropy is nearly Ising with a strong tendency for the moments to align close to the crystallographic $c$-axis. **f** Magnetization vs applied magnetic field for polycrystalline $Yb_2Be_2SiO_7$ at 2 K, 5 K, and 10 K. The measured values at 1.5 T are labeled on the panel.

Fig. 1b. The local $y$-axes are parallel to the mirror-plane normals and perpendicular to the intradimer bonds. One principal g-tensor direction lies along the local $y$-axis. The other two principal g-tensor directions are not constrained by symmetry and make an angle of $\beta$ with the crystallographic [001] and [110] or [1$\bar{1}$0] directions. The local coordinate system sketched in Fig. 1b applies when $\beta = 0$. Under this assumption, the blue arrows in this same panel denote the local $x$-axes, which are parallel to the dimer bonds.

Due to the low point symmetry of the $Yb^{3+}$ ions and the lack of sufficient crystal field observables in the neutron spectroscopy data, it is not feasible to obtain a unique set of crystal field parameters for $Yb_2Be_2SiO_7$. Half-polarized neutron powder diffraction (pNPD) offers a powerful alternative to probe the single-ion anisotropy of this system via the local site susceptibility approach[55]. pNPD is performed in the paramagnetic region and therefore the lack of long-range order does not influence the application of this technique. Spin-up and spin-down neutron diffraction patterns with intensities of $I^+$ and $I^-$ are measured separately, and then Rietveld refinements of the sum and difference patterns (given by $I^+ + I^-$ and $I^+ - I^-$, respectively) are performed using the software CrysPy[55]. The $Yb^{3+}$ local site susceptibility tensor, which provides information on the local anisotropy and principal g-tensor directions and their relative magnitudes, can be extracted from the difference pattern.

The difference pattern $I^+ - I^-$ measured at 5 K in an applied magnetic field of 1.5 T is shown in Fig. 2d and the sum pattern from the same dataset is shown in Supplementary Fig. S2. In each case, the Rietveld refinement result is superimposed on the data and the fit residual is plotted below the data. The best-fit result of the difference pattern yields a small $\beta$ value of 1.52(3)°, which means that the two principal g-tensor directions not constrained by symmetry are nearly aligned with the crystallographic [001]-axis and the dimer bond

direction ([110] and [1$\bar{1}$0] for the two dimer sublattices) as illustrated in Fig. 1b. The magnetization ellipsoids' principal axes have magnitudes of 2.32 $\mu_B$/T, 0.14 $\mu_B$/T, and 0.10 $\mu_B$/T, with the largest value for the pseudo-[001]-axis direction. These results are indicative of an Ising-like anisotropy with small deviations away from the [001]-axis and they are consistent with our Curie–Weiss fitting results described above. The analysis of the 10 K difference pattern measured in the same 1.5 T magnetic field produces similar results. The magnetization ellipsoids calculated from the local site susceptibility analysis are presented in Fig. 2e. CrysPy can also be used to calculate the powder-averaged magnetization from the local-site susceptibility tensor to facilitate a direct comparison to the experimental data as a sanity check. The powder-averaged magnetization at selected temperatures for $Yb_2Be_2SiO_7$ is presented in Fig. 2f with the measured 1.5 T values shown on the panel. The 5 K and 10 K values calculated from the refined local site susceptibility tensors are 0.53 $\mu_B$/Yb and 0.29 $\mu_B$/Yb, respectively, which are nearly the same values from our powder-averaged magnetization measurements.

## Isolated dimer model

Low-energy neutron powder spectroscopy data were used to investigate the magnetic excitations associated with the collective ground state of $Yb_2Be_2SiO_7$. A color contour plot of the scattering intensity as a function of $Q$ and $E$ with $T = 0.25$ K measured on CNCS with $E_i = 1.55$ meV is presented in Fig. 3a. A simulated scattering pattern based on an isolated dimer model that will be discussed below with Gaussian peak widths (full-width half maximums) that match the expected instrumental energy resolution at the mode positions (i.e. 0.035 meV at 0.11 meV and 0.033 meV at 0.19 meV) is depicted in Fig. 3b. A low-energy band of excitations with minimal dispersion centered at 0.11 meV and an intensity maximum as $Q \to 0$ is visible in

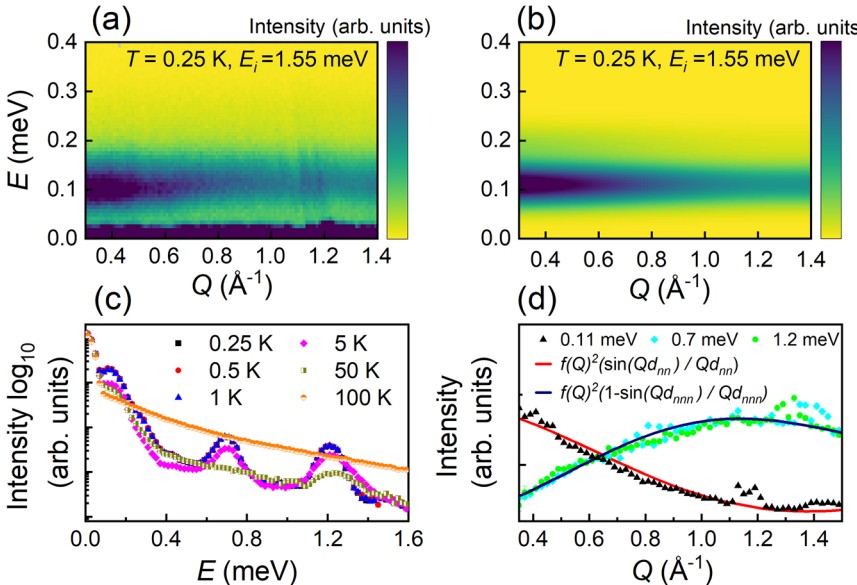

**Fig. 3 | Dynamical structure factor. a** Color contour plot of the scattering intensity as a function of momentum and energy transfer at 0.25 K from CNCS with $E_i = 1.55$ meV. A weakly-dispersive mode with a magnetic origin is centered at 0.11 meV. **b** Color contour plot of the simulated scattering intensity using an isolated dimer model with the anisotropic intradimer exchange parameters provided in the main text. **c** Constant-$Q$ cuts of the scattering intensity ($Q$ integration range [0.5, 1.8] Å$^{-1}$) with $E_i = 2.49$ meV at selected temperatures. Aside from the main excitation band centered at 0.11 meV with a shoulder at 0.19 meV, there are two higher excitation bands centered at 0.7 meV and 1.2 meV, respectively. **d** Constant-$E$ cuts centered about three different mode positions, with an energy integration range ±0.05 meV for the lowest mode ($E_i = 1.55$ meV data) and an integration range of ±0.1 meV for the two higher modes ($E_i = 2.49$ meV data). The intensities of the two higher-energy cuts have been rescaled to facilitate a straightforward comparison of their $Q$-dependence, which is strikingly different from the lower excitation band. Simulations based on two different types of single-dimer ground states are superimposed on the data and describe it well.

the data. There is also a second excitation at 0.19 meV that manifests as a shoulder in the data. Both mode energies were determined by fitting the data to a sum of Gaussian functions as shown in Supplementary Fig. S3 and summarized in Supplementary Table S4. Constant-$Q$ cuts at selected temperatures are presented in Supplementary Fig. S4 for the $E_i = 1.55$ meV dataset and over a wider energy range in Fig. 3c for the $E_i = 2.49$ meV dataset. The logarithmic intensity scales help to reveal two higher-energy excitation bands centered at 0.7 meV and 1.2 meV. The $Q$-dependence of the three distinct modes is shown in Fig. 3d. Both the $Q$ and $T$-dependence of these modes indicate that they are magnetic excitations and their nearly-dispersionless nature suggests that they are associated with isolated or weakly-interacting spin dimers.

Since effective spin-1/2 dimer models with strong intradimer exchange anisotropy can have a maximum of three single-dimer excitations, it is clear that all four modes (at 0.11, 0.19, 0.7 and 1.2 meV) observed here do not have the same origin. We first verified that none of these modes were spurious by measuring them with different incident energies. Zero-field heat capacity only depends on the eigenvalues of these models, so we compared our data shown in Fig. 2a to simulations assuming various energy-level schemes (see Supplementary Fig. S5). The best simulation of the zero-field heat capacity data corresponds to an isolated dimer model with a doubly-degenerate mode at 0.11 meV and a non-degenerate mode at 0.19 meV. This implies that the two higher-energy modes have a different origin. In fact, as shown in Fig. 3d their $Q$-dependence is well-described by the Yb$^{3+}$ magnetic form factor squared multiplied by the known structure factor of a triplet excitation for an isolated Heisenberg dimer model[56] with the square plaquette distance $d_{nnn} = 3.84$ Å. Since these higher-energy modes have a negligible contribution to the specific heat, this implies that only a small fraction of Yb$^{3+}$ ions form longer-range dimers in Yb$_2$Be$_2$SiO$_7$, possibly due to the known Be/Si site mixing in this system. We also compared the $Q$-dependence of these modes to expectations for the lowest-energy excitations of rectangular

tetramers[56], but this model could not account for the $Q$-position of their maximum intensities. Finally, we note that higher-energy excitations in SrCu$_2$(BO$_3$)$_2$ have been attributed to singlet bound states consisting of contributions from multiple dimer units[57–59] and a similar alternative origin for the higher-energy modes cannot be ruled out in the present case.

With the energy level scheme for a possible isolated dimer model comprised of most Yb$^{3+}$ ions in the system established, we performed a quantitative analysis of the low-energy neutron spectroscopy data and field-dependent heat capacity and magnetization data. We started by constructing a minimal model based on symmetry analysis[46,60]. The model Hamiltonian describes a collection of isolated dimers with anisotropic intradimer interactions and Zeeman coupling:

$$\mathcal{H} = \sum_{<i,j> l\alpha} S_{il}^\alpha J_{\alpha\alpha}^l S_{jl}^\alpha - \mu_B \sum_{il\alpha\beta} H^\alpha g_{\alpha\beta}^l S_{il}^\beta \tag{1}$$

where the superscript $l$ denotes dimers in sub-lattice A/B, $S_i^\alpha$ refers to the $\alpha(= x, y, z)$ component of the effective spin 1/2 operator with $x = [1, 1, 0]$, $y = [-1, 1, 0]$ and $z = [0, 0, 1]$, $J_{\alpha\alpha}^l$ represents one component of the intradimer exchange tensor for dimers in each sub-lattice, $H^\alpha$ is the $\alpha$ component of the applied magnetic field, and $g_{\alpha\beta}^l$ represents one component of the $g$-tensor. Our model is defined in global crystallographic coordinates and it is equivalent to the one defined within the local basis after proper rotations. Constrained by the time-reversal and mirror symmetries, this Hamiltonian takes a diagonal form with an $XYZ$ type of spin anisotropy in the reference frame defined above, and the couplings on the two sub-lattices are related by a 90° rotation about the crystallographic [001]-axis so that $J_{xx/yy}^A = J_{yy/xx}^B$ and $J_{zz}^A = J_{zz}^B$. Off-diagonal $g$-tensor components are allowed by symmetry, but our pNPD results described above suggest they are much smaller than the diagonal ones. We hence take the approximation that the g-tensor follows the same symmetry as the exchange tensor. With

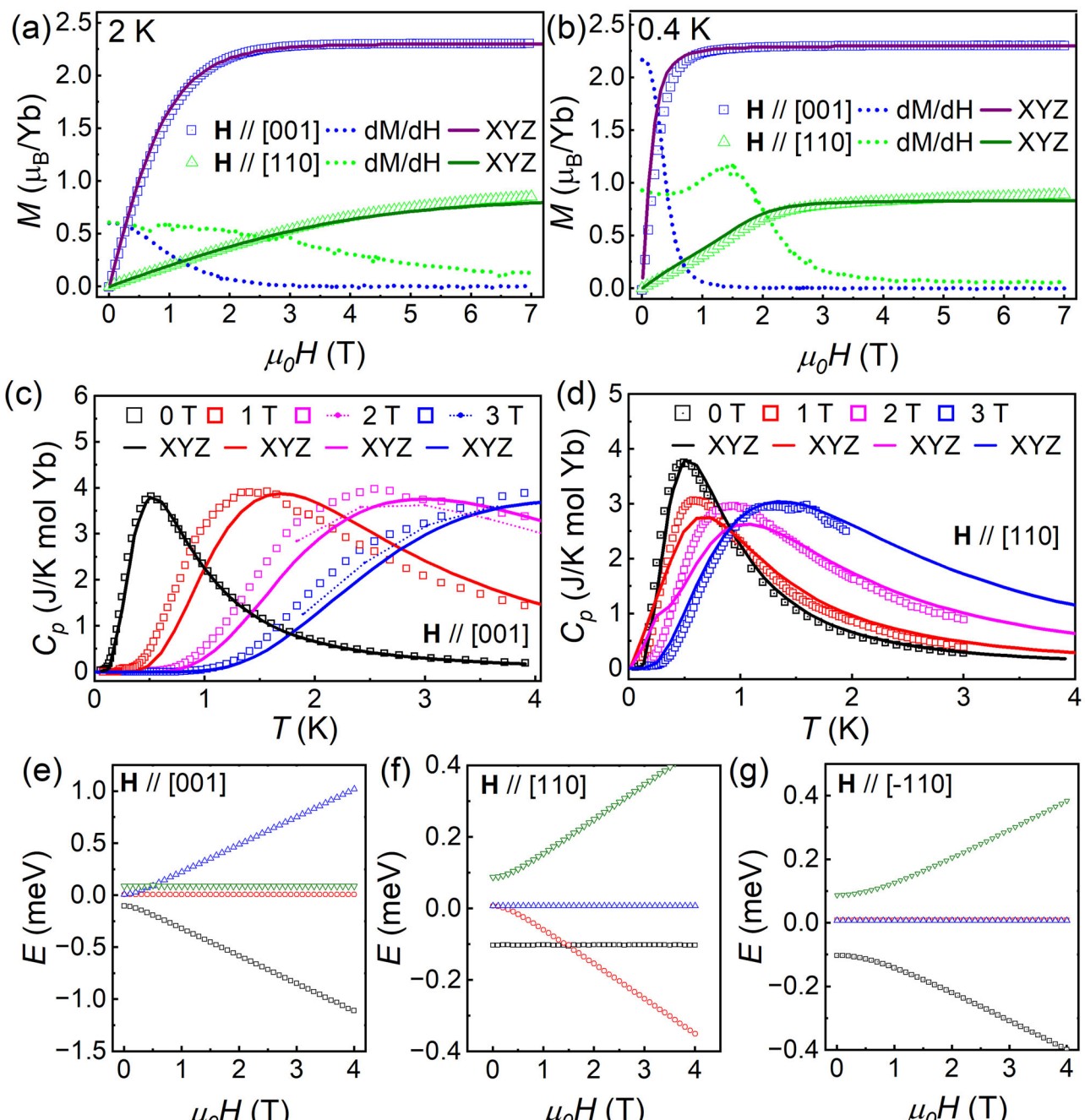

**Fig. 4 | Isolated dimer modeling. a** Magnetization vs field along two high-symmetry crystallographic directions at 2 K. **b** Similar data collected at 0.4 K. **c** Heat capacity vs magnetic field applied along the [001] direction. The initial experimental data with a modest field misalignment are shown as open symbols, the remeasured 2 T and 3 T data over a limited temperature range are shown as points connected by dashed curves, and the simulated data from our XYZ anisotropic exchange model are shown as solid curves. **d** Heat capacity data vs magnetic field applied along the [110] direction with simulated data from the same model superimposed on it. **e–g** Simulated isolated dimer energy levels vs applied field along different crystallographic directions for one dimer sublattice based on the XYZ anisotropic exchange model. There is a level crossing near 1.5 T for one dimer sublattice when **H**‖[110] or for the other dimer sublattice when **H** ‖ [$\bar{1}$10].

the principal $g$-tensor directions now fixed along high-symmetry crystallographic directions, we estimated the $g$-tensor values by using the 2 K magnetization data, with the **H**‖[001] and **H**‖[110] measurements presented in Fig. 4a. After subtracting off a linear van-Vleck contribution in the highest-field regime, we obtain $g_{xx}^A = 1.64$, $g_{yy}^A = 1.71$, and $g_{zz}^A = 4.6$.

The energy level scheme established from the neutron spectroscopy and zero-field heat capacity data described above is consistent with 12 different intradimer exchange matrices, which we determined using Supplementary Eq. (S2). We then performed exact

diagonalization of these 12 Hamiltonians in zero field to find six solutions with $S_z = 0$ ground state wavefunctions and six solutions with $S_z \neq 0$ ground state wavefunctions. We calculated the dimer structure factors for the three excitations of each model using Supplementary Eq. (S4), with the results for three representative models shown in Supplementary Fig. S6. Importantly, all 12 models have one intense mode and two much weaker modes. This common feature allows us to reliably compare the $Q$-dependence of the intense mode for the models with the neutron spectroscopy data even though neither weak mode is completely resolved from the intense one. The two types of

solutions are characterized by intense dimer excitations with very different $Q$-dependences. In the regime $Q \leq 1.6$ Å$^{-1}$, the intense excitation for the $S_z = 0$ solutions is described by the structure factor,

$$S(Q) = A\left(1 - \frac{sin(Qd)}{Qd}\right) \qquad (2)$$

where $A$ is a constant and $d$ is the intradimer distance. In the same $Q$-regime, the intense excitation for the $S_z \neq 0$ solutions is described approximately by the structure factor

$$S(Q) \approx A\frac{sin(Qd)}{Qd}. \qquad (3)$$

To narrow down possible intradimer exchange matrices, we examined the $Q$-dependence of the constant $E$-cut centered at the intense 0.11 meV mode with an integration range of ±0.05 meV shown in Fig. 3d. The data fits well to the expected structure factor for the intense mode of the six $S_z \neq 0$ solutions. Two of these options can be ruled out because their most intense mode is located at 0.19 meV, which is inconsistent with the neutron spectroscopy data. This leaves four viable solutions and the corresponding simulations for them are presented in Supplementary Fig. S7.

All four remaining candidate models have single-dimer ground states of the form $(|\uparrow\uparrow\rangle + |\downarrow\downarrow\rangle)/\sqrt{2}$ or $(|\uparrow\uparrow\rangle - |\downarrow\downarrow\rangle)/\sqrt{2}$ with a dominant contribution from the other $S_z \neq 0$ wavefunction to the doubly-degenerate first excited state. We note that these two dimer ground states are qualitatively consistent with the low-$T$ magnetic susceptibility data presented in Fig. 1d, which should level off at finite values for all field orientations as $T \to 0$ rather than drop to zero as expected for a Heisenberg dimer model. Two of the solutions show superior agreement with the bulk characterization data. Although the $Q$-dependence of the weak mode intensities from neutron powder spectroscopy is not exactly the same for these two models, our data is not sufficient for differentiating between them conclusively. The exchange parameters for one of these models are $J_{xx} = 0.19$ meV, $J_{yy} = -0.03$ meV, and $J_{zz} = -0.19$ meV and the exchange parameters for the second model are $J_{xx} = -0.03$ meV, $J_{yy} = 0.19$ meV, and $J_{zz} = -0.19$ meV.

Using the Hamiltonian parameters provided above for the first model, the neutron spectroscopy simulation is shown in Fig. 3b and the calculated single crystal magnetization and heat capacity data are superimposed on the corresponding experimental datasets in Fig. 4a–d. It is worth noting that there is no evidence for field-induced magnetic order in the heat capacity measurements down to dilution fridge temperatures. The agreement between experiment and theory is excellent, with the small differences between the $\mathbf{H}\|[001]$ heat capacity data and the simulation arising from modest sample misalignment that was confirmed by re-measuring the 2 T and 3 T data with coarse temperature steps for $T \geq 1.8$ K, which are shown as points connected by dashed curves in Fig. 4c. This simulation yields an effective $S = 1$ dimer state $(|\uparrow\uparrow\rangle - |\downarrow\downarrow\rangle)/\sqrt{2}$ in zero field, which is stabilized by the large ferromagnetic exchange interaction $J_{zz}$ along the quantization axis. The excited states are doubly degenerate $(|\uparrow\uparrow\rangle + |\downarrow\downarrow\rangle)/\sqrt{2}$ and $(|\uparrow\downarrow\rangle - |\downarrow\uparrow\rangle)/\sqrt{2}$ at $E = 0.11$ meV and $(|\uparrow\downarrow\rangle + |\downarrow\uparrow\rangle)/\sqrt{2}$ at $E = 0.19$ meV, respectively. The second model has a similar eigenvector scheme for its modes, although the two $S_z \neq 0$ modes swap their eigenvalues so the ground state is $(|\uparrow\uparrow\rangle + |\downarrow\downarrow\rangle)/\sqrt{2}$ instead. Both isolated dimer models account for the major experimental features well, including the $Q$ and $E$-dependence of the dynamical structure factor for the dimer excitations and the temperature and magnetic-field-dependence of the magnetization and specific heat. These results confirm that the interdimer interactions are significantly weaker than the intradimer interactions in Yb$_2$Be$_2$SiO$_7$.

Beyond the qualitative signatures of the $S_z \neq 0$ dimer ground state in neutron spectroscopy, we also observe an anisotropic magnetization process that is not expected for systems with $S_z = 0$ dimer ground states. In particular, no low-field magnetization plateaus are observed at 0.4 K when the magnetic field is applied along any high-symmetry crystallographic direction, which is in sharp contrast to expectations for $S_z = 0$ dimer ground states as shown in Supplementary Fig. S6. Instead, applying a magnetic field along the [001] direction continuously rotates the $(|\uparrow\uparrow\rangle - |\downarrow\downarrow\rangle)/\sqrt{2}$ ground state without closing the excitation gap so that the system is immediately magnetized and evolves to the fully-polarized state adiabatically without a transition as shown in the Fig. 4b data and illustrated by the energy level vs field diagram presented in Fig. 4e. On the other hand, applying the field along the [110] direction causes a level crossing for one dimer sublattice as shown in Fig. 4f, which corresponds to a quantum phase transition to an intermediate state as indicated by the peak in $dM/dH$ shown in Fig. 4b. The other dimer sublattice has a similar level crossing when $\mathbf{H} \| [1\bar{1}0]$. It is worth noting that the largest discrepancy in the $\mathbf{H}\|[110]$ bulk characterization data and the simulations is observed in the fixed-field heat capacity measurements shown in Fig. 4d near the expected critical field for the level crossing. This is likely due to neglecting the appropriate sub-leading interaction in the simulations, which should have the largest effect on the magnetic properties of the system in this regime[46]. Indeed, neutron powder spectroscopy also suggests that a small additional term should be added to the Hamiltonian to capture the increased bandwidth of the 0.11 meV mode observed in the data, but its accurate determination without single-crystal neutron spectroscopy data is challenging. The nature of this term is not obvious either, as the expected sub-1 K energy scale means it could arise from interdimer exchange or a dipolar interaction. While additional theoretical work on candidate magnetic Hamiltonians for this system and its isostructural family members could help to address this issue, neglecting this extra term does not affect the exotic zero-field magnetic ground state for Yb$_2$Be$_2$SiO$_7$ established here.

## Discussion

Quantum dimer magnets with pure spin-1/2 degrees of freedom have been studied extensively[4,8]. Antiferromagnetic Heisenberg exchange at the intradimer level stabilizes the entangled singlet ground state $(|\uparrow\downarrow\rangle - |\downarrow\uparrow\rangle)/\sqrt{2}$ in zero field with a spin gap to a triplet excited state. Key experimental signatures of this exotic ground state are an isotropic decrease in the magnetic susceptibility with decreasing temperature and zero magnetization (i.e., a magnetization plateau) at sufficiently low temperatures and applied fields, along with a powder-averaged dynamical structure factor for the triplet excitation that satisfies Eq. (2) exactly[56]. Quantum dimer magnets with effective spin-1/2 degrees of freedom can exhibit very different behavior due to their realization of anisotropic exchange Hamiltonians. The type of magnetic ion and its point symmetry, along with the composition of the crystal-field ground-state wavefunctions, can give rise to intradimer exchange matrices with different symmetries including Ising, XY, XXZ, or even XYZ. The quasi-orthorhombic point symmetry for the Yb$^{3+}$ ions in Yb$_2$Be$_2$SiO$_7$ generates the XYZ dimer model, which was only recently explored by theorists[60]. Their most important insight was that the ground state of the XYZ model can be any one of the four possible dimer states depending on the sign and relative strength of the three anisotropic exchange matrix terms. One can distinguish between the four dimer ground states via qualitatively different magnetic susceptibilities, magnetizations, and powder-averaged dynamical structure factors, as shown in Supplementary Fig. S6. We have measured all three quantities for Yb$_2$Be$_2$SiO$_7$ and found that their behavior matches expectations for the novel $(|\uparrow\uparrow\rangle - |\downarrow\downarrow\rangle)/\sqrt{2}$ or $(|\uparrow\uparrow\rangle + |\downarrow\downarrow\rangle)/\sqrt{2}$ ground state.

Our results have broad implications for other quantum dimer magnets with effective spin-1/2 degrees of freedom and show that

anisotropic intradimer exchange should always be considered when trying to understand the magnetic properties of these materials. $Yb_2Si_2O_7$ was the first insulating rare-earth-based quantum dimer magnet discovered with effective spin-1/2 moments[61]. While the bulk characterization and neutron scattering results are broadly consistent with a $(|{\uparrow\downarrow}\rangle - |{\downarrow\uparrow}\rangle)/\sqrt{2}$ zero-field ground state, the magnetic Hamiltonian is still unknown. An interacting version of the XYZ dimer model discussed here may be appropriate due to the low-point symmetry of the Yb ions in that system. Anisotropic intradimer exchange should also be important for the new insulating SSL families $BaR_2MX_5$ ($R$ = rare earth; $M$ = Zn, Pd, or Pt; $X$ = O or S) and $R_2Be_2ZO_7$ ($Z$ = Ge or Si). In fact, it has been shown to stabilize the same $(|{\uparrow\uparrow}\rangle - |{\downarrow\downarrow}\rangle)/\sqrt{2}$ dimer ground state in the effective spin-1/2 system $BaCe_2ZnS_5$[46] that we have identified as one possibility for $Yb_2Be_2SiO_7$ here. Despite this progress, direct links between models with anisotropic intradimer exchange and the measured magnetic properties of these materials have remained scarce. There is now ample materials' phase space to explore this topic in more detail.

In summary, we have characterized the low-temperature magnetic properties of the quantum dimer magnet $Yb_2Be_2SiO_7$ using bulk characterization and neutron scattering techniques. We find that the effective spin-1/2 $Yb^{3+}$ moments form an unconventional, bipartite entangled state arising from dominant, anisotropic intradimer exchange. The magnetic Hamiltonian for $Yb_2Be_2SiO_7$ gives rise to a dimer ground state with a $S_z \neq 0$ ground state wavefunction instead of the $(|{\uparrow\downarrow}\rangle - |{\downarrow\uparrow}\rangle)/\sqrt{2}$ state commonly realized in Heisenberg dimer systems based on spin-1/2 moments. A significant amount of Be/Si site mixing in $Yb_2Be_2SiO_7$ may give rise to the broad crystal field levels and possibly the two higher-energy dimer excitations. Our work shows that quantum dimer magnets with strong spin-orbit coupling are promising playgrounds for identifying unusual entangled states of matter in quantum materials.

## Methods

### Sample preparation
Polycrystalline samples of $Yb_2Be_2SiO_7$ were prepared from stoichiometric amounts of $Yb_2O_3$, BeO, and $SiO_2$ as detailed in ref. 48. Small single crystals were grown using the floating zone melt method[62,63] and oriented via Laue back-diffraction[64,65]. Detailed structure refinements of single-crystal x-ray diffraction data (XRD) agree well with the previously-reported structure obtained from powder XRD[48] and the results are provided in Supplementary Tables S1, S2.

### Bulk characterization
Polycrystalline and single-crystal DC magnetization measurements as a function of magnetic field and temperature were performed using an MPMS3 SQUID magnetometer (Quantum Design) equipped with a He-3 insert. For the single crystal measurements, the magnetic fields were applied along high-symmetry crystallographic directions. Specific heat measurements using single crystals were performed with a Quantum Design Physical Property Measurement System (PPMS) equipped with a dilution refrigerator insert. The AC susceptibility measurements were conducted on single crystals with a voltage-controlled current source (Stanford Research, CS580) and lock-in amplifier (Stanford Research, SR830). The phase of the lock-in amplifier was set to measure the first harmonic signal. The RMS amplitude of the AC excitation field was set to be 0.6 Oe with the frequency fixed to be 200 Hz. The measurements were performed in the SCM1 Dilution Refrigerator of the National High Magnetic Field Laboratory, Tallahassee.

### Neutron scattering
Neutron powder diffraction (NPD) was performed using the high-resolution HB-2A powder diffractometer[66,67] at the High Flux Isotope Reactor (HFIR) of Oak Ridge National Laboratory (ORNL). Unpolarized measurements were conducted with a ~4 g polycrystalline sample sealed inside an aluminum can with 1 atm of helium exchange gas. Diffraction patterns were collected at 250 mK and 2 K with a wavelength of 2.41 Å and a collimation of open-open-12'.

A half-polarized neutron powder diffraction (pNPD) experiment was performed using the same HB-2A collimation settings and a V-cavity to generate the polarized beam with a ~13 g polycrystalline sample pressed into pellets. The sample was loaded in a larger Al can than the one used for the unpolarized measurements. The polarization state was controlled using a Mezei flipper. Further details of the HB-2A pNPD experimental set-up can be found in ref. 68. Spin-up and spin-down diffraction patterns, with intensities denoted by $I^+$ and $I^-$ respectively, were collected in a vertical magnetic field of 1.5 T at temperatures of 5 and 10 K using a wavelength of 2.41 Å.

Time-of-flight neutron powder diffraction data were collected at 100 K on NOMAD[67] at the Spallation Neutron Source (SNS) of ORNL using ~4 g of powder. Similar measurements were carried out on ~4.5 g of crushed single crystals and ~2.3 g of polycrystalline $Er_2Be_2SiO_7$[51].

Neutron powder spectroscopy data were obtained from the direct-geometry time-of-flight instrument SEQUOIA[69] at the SNS using ~2.5 g of similar $Yb_2Be_2SiO_7$ powder. The same amount of a non-magnetic reference sample $Lu_2Be_2SiO_7$ was also measured. All SEQUOIA data were collected at 5 K with an incident energy of 80 meV using the fine Fermi chopper. The $T_0$ frequency, Fermi chopper frequency, and energy resolution at the elastic line (full-width half-maximum) were 90 Hz, 480 Hz, and 2.15 meV.

Lower-incident-energy neutron powder spectroscopy data were measured with the direct-geometry time-of-flight instrument CNCS[70] at the SNS using the same polycrystalline sample from the unpolarized HB-2A experiment. The CNCS data were collected using incident energies of $E_i$ = 1.55 meV and 2.49 meV in the "high flux" chopper setting mode, which produced energy resolutions of 0.04 meV and 0.06 meV (full-width half-maximum) at the elastic line, respectively.

## Data availability
All the data supporting the findings of this study are available within the article and from the corresponding authors upon request. Source data are provided with this paper.

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

## Acknowledgements

Research at the University of Tennessee is supported by the National Science Foundation, Division of Materials Research under Award No. NSF-DMR-2003117. The work at Michigan State University is supported by the U.S.DOE-BES under Contract DE-SC0023648. The work performed at NHMFL is supported by the NSF Cooperative Agreement No. DMR-1644779 and the State of Florida. A portion of this research used resources at the Spallation Neutron Source and the High Flux Isotope Reactor, which are DOE Office of Science User Facilities operated by Oak Ridge National Laboratory. The beam time was allocated to HB-2A (POWDER) on proposal number IPTS-29325.1 and IPTS-33066.1. Additional beam time was allocated to BL-1B (NOMAD) on proposal number IPTS-31286.1, BL-5 (CNCS) on proposal number IPTS-30936.1, and BL-17 (SEQUOIA) on IPTS-29415.1. A portion of the research at ORNL was supported by the DOE, Office of Science, Office of Advanced Scientific Computing Research (contract No. ERKJ387), and Office of Basic Energy Sciences (award No. KC0402020 under contract No. DE-AC05-00OR22725). Work by B.A.F. (pair distribution function analysis) was supported by the National Science Foundation LEAPS-MPS program through Grant No. NSF-DMR-2418438. G.D., R.Y., N.L., and X.F.S. thank the support from the National Key R&D Program of China (Grant No. 2023YFA1406500). G.D. and R.Y. were also supported by the National Natural Science Foundation of China (Grant Nos. 12334008 and 12174441). N.L. and X.F.S. were also supported by the National Natural Science Foundation of China (Grant Nos. 12274388 and 12404043) and the Nature Science Foundation of Anhui Province (Grant No. 2408085QA024). C.L. was supported by the National Natural Science Foundation of China (Grant No. 12564021).

## Author contributions

A.B., H.D.Z., and A.A.A. conceived the project. A.B., H.D.Z., N.L., X.F.S., and E.S.C. synthesized the samples and performed the bulk characterization measurements. H.W. and W.X. collected and analyzed the single crystal x-ray diffraction data. A.B., Q.M., S.C., A.I.K., K.M.T., G.S. and A.A.A. participated in the neutron scattering data collection. A.B., Q.M., G.D., S.C., B.A.F., C.L., R.Y., H.D.Z. and A.A.A. contributed to the remaining data analysis. The manuscript was written by A.B. and A.A.A. Critical manuscript comments were provided by all authors.

## Competing interests

The authors declare no competing interests.
