## [Peer Review File · Nature Communications]

Unconventional bipartite entanglement in the quantum dimer magnet $\text{Yb}_2\text{Be}_2\text{SiO}_7$

Corresponding Author: Dr Adam Aczel

Version 0:

Reviewer comments:

Reviewer #1

(Remarks to the Author)

Comments:

A Brassington et al reported the possible realization of novel entangled spin dimer state in Shastry-Sutherland lattice $\text{Yb}_2\text{Be}_2\text{SiO}_7$ compound. This class of materials is interesting for investigating the anisotropic spin dimer physics beyond isotropic Heisenberg model. The purported conclusion is that the ground states of $\text{Yb}_2\text{Be}_2\text{SiO}_7$ can be described by an isolated dimer model with highly anisotropic exchanges, which is mainly based on the theoretical model but no typical experimental signatures observed in specific heat, magnetization and neutron scattering. The experimental evidence on the formation of novel singlet state is quite weak and can't not support the conclusion, thus it is not reach the level of impact and not enough for publication on Nat. Commun. The comments are listed as below for consideration:

1) $\text{Yb}_2\text{Be}_2\text{SiO}_7$ possibly realizes the anisotropic spin dimer physics beyond isotropic Heisenberg model. But, the observed anisotropic magnetization behaviors can't be simply ascribed to the anisotropic exchange interactions since it can be related to the single ion effect of Yb^{3+} ions because similar anisotropy is observed at 2 K far above the energy scale of exchange interactions. Since broad peak centered at 0.5 K, the anisotropic magnetization characterization far below 0.5 K is necessary to check whether this is field-induced phase transition, that can identify whether the system is isolated spin dimer or interacted spin dimer. Without this characterization, the authors can't give the conclusion this is isolated system and exclude the $\frac{1}{\sqrt{2}}(|\uparrow\downarrow\rangle - |\downarrow\uparrow\rangle)$ ground states. Also, in the specific heat data, no experimental data is provided at the field regions between 0-1T, which can give some information on CEF level crossing for spin dimer. We proposed the authors to characterize the related properties.

2) If the ground state is $|\psi_0\rangle = \frac{1}{\sqrt{2}}(|\uparrow\uparrow\rangle - |\downarrow\downarrow\rangle)$ singlet state, the magnetic susceptibility should drop down to zero value as temperature approaches zero Kelvin. How about the susceptibility data? The size of energy gap between the singlet state and first excited state?

3) For the Neutron scattering of $\text{Yb}_2\text{Be}_2\text{SiO}_7$, the powder samples are used that cannot give the dispersion relation of spin excitation, as presented in Figs. 2 and 3. The experimental data is enough to determine the CEF levels, the resolution is not high enough that the dispersionless nature of excitation can not be concluded based on the present data. And, how to understand the double mode at 0.11 meV and 0.19 meV, the related experimental measurements on single crystal is appreciated to clarify this experimental behavior. Only in that case, the excite states can be fully confirmed alongside their proposed model.

4) The spin dimer physics in Ce/Yb based SSL compounds included the proposed singlet state have been discussed in the similar compounds $\text{BaCe}_2\text{ZnS}_5$ [arXiv:2412.17913,2024] and $\text{Yb}_2\text{Be}_2\text{GeO}_7$ [Phys. Rev. B 110, 144445 (2024)]. The similar claim is reported, the related work should be cited.

5) In Figure S7 of supporting information, we don't observe the double broad peaks in the specific heat near critical field for closing the gap of spin dimer. While, the proposed theoretical model yield the two humps in $C_p(T)$ curves, this contradicts with the formation of isolated spin dimer, how to explain this controversy.

Reviewer #2

(Remarks to the Author)

Novel entangled dimer state in the Shastry-Sutherland magnet $\text{Yb}_2\text{Be}_2\text{SiO}_7$

The authors present a thorough experimental investigation of the material Yb₂Be₂SiO₇ by using a combination of low-temperature neutron scattering, bulk magnetometry and specific heat measurements. It is argued in the paper that the material realizes an anisotropic variant of the Shastry-Sutherland model (SSM) in the weakly-interacting interdimer regime. The SSM is a quintessential model in quantum magnetism and the results presented in the paper, if correct, would cement Yb₂Be₂SiO₇ as a novel material in which anisotropic SSM physics could be further explored in e.g. future single-crystal inelastic neutron scattering (INS) studies. Although the agreement between the model calculations and the experimental data is overall excellent and demonstrates a deep and accurate interpretation of the physics of Yb₂Be₂SiO₇, we find that the quality of the presentation of these results has room for improvement in order to be understandable by a broad audience not familiar with the plethora of neutron scattering techniques used in the paper. In addition, the potential impact that these new results have, could be motivated further, especially given the wealth of literature on SSM physics and other recent experimental efforts on crystal systems featuring SSM physics. For these general reasons we suggest that the authors rework the paper before we can support it for publication.

On the results:

-Abstract: Revise the sentence "a singlet ground state with the symmetric wavefunction"

-Introduction and conclusions: link results to existing theoretical work

Page 5: It is not stated in the text how the inelastic neutron scattering data (proportional to $S(Q,E)$) is compared to equations (2) and (3). Is the partially Q-integrated powder data integrated along E in order to yield the total scattering intensity $S(Q)$? In Ref. [48] the term "exclusive structure factor" is used to refer to the Q-dependence of $S(Q,E)$ at a fixed E corresponding to a selected transition from the ground state. Is this definition adopted in the paper? If so it should be made explicit.

Equation (2) is derived in Ref. [48]. The authors do not derive explicitly nor provide a reference that leads to a Q-dependence of the form shown in Eq. (3) (Eq. (6) in the SM). Is this equation rather a phenomenological structure factor that goes to a constant value as $Q \rightarrow 0$ as seen in experiment?

Is there a physical reason why the intradimer xx-exchange is ferromagnetic whereas the zz-exchange is antiferromagnetic? Could the sign of the exchange tensor elements be determined from the data that entered the fit? In other words, did the fit procedure allow to determine whether the exchange is AF or AFM?

A discussion on the results put in the context of Refs [39, 49 and 51] would be beneficial. Especially given that the ground state wavefunction of Yb₂Be₂SiO₇ deduced from the heat capacity and INS data appears identical to the one found by the authors of Ref. [39] for their model system.

Page 4: specify quantitatively how good the agreement is between the measured polycrystalline magnetization and the calculated powder-averaged magnetization.

On the presentation:

Discussion regarding the g-tensor in page 2 can be left for after the pNDP results and discussion.

Fig.2(f) clarify meaning of values (0.28,0.52,1.03) in figure caption.

As measurements were made both in powder and single crystal samples, specify the sample form for the results in the main text, eg. "Powder neutron spectroscopy" instead of "Neutron spectroscopy"

Specify instrumental energy resolution for/or in Fig.2(c)

For the published version, the paper would benefit greatly from higher-quality (higher-DPI) figures throughout the text. The xyz coordinate axes in Fig. 1 for example are barely legible in a printed version of the paper.

Supplement, section II. The equation relating the local susceptibility tensor to the magnetization and applied field should be written in component form to avoid a notation that implies division by a vector.

Supplement, figure S6. The x-label for panels (a), (c) and (e) should read Q (\AA^{-1}).

Reviewer #3

(Remarks to the Author)

The authors have presented a comprehensive study of the magnet Yb₂Be₂SiO₇ with a Shastry-Sutherland lattice structure. The main finding of the manuscript is that this compound can be described well by isolated dimers where the ground state of each dimer is an effective $S=1$ state ($|\uparrow\uparrow\rangle - |\downarrow\downarrow\rangle$)/ $\sqrt{2}$. This conclusion is firmly established via thermodynamic measurements, inelastic neutron scattering (to obtain CEF scheme), half-polarized neutron diffraction (to obtain the magnetic anisotropy), and a model using isolated dimers. In particular, I am quite impressed by the authors' effort of using the half-polarized neutron powder diffraction to clarify the magnetic anisotropy of the compound, which is a crucial step for accurately determining the model. Overall, this manuscript appears to be a very solid work.

However, I am not in a positive position in supporting the publication of the manuscript in Nature Communications, at least not in the current form. In fact, in contradiction to what the authors have claimed in the title, the results of this manuscript show that this compound has "boring disentangled" dimer state, and despite the lattice structure the physics of this material has almost NOTHING to do with "Shastry-Sutherland" physics. As the authors have successfully demonstrated, the physics of this material is a simple "molecular"-like paramagnet, which can be described by isolated (disentangled) dimers (molecules).

I believe that both the authors and the editor would agree with me that reporting negative results is also a helpful way in maintaining the healthy growth of the scientific community. I understand that the authors were trying to find a fancy title so the manuscript could have a higher chance in glossy journals like the Nature series. However, in this particular case, it seems to me it does not hurt to publish this paper with an unbiased title in Nature Communications. Due to the increasing interest in finding novel Shastry-Sutherland physics in rare-earth based systems, this paper could save the global research community working on related compounds a lot of money/time, by simply letting people know that there is no Shastry-Sutherland physics in this particular compound (and likely also in sibling compounds). In this sense, publishing this paper with an

unbiased message in Nature Communications will be greatly helpful to the community. My suggestion is to modify the title and the discussions of the paper to unbiasedly reflect the physics of this material, although it is in some sense a “negative” result.

Another minor note: I am confused by comparing Fig. 4(f) with $H//[-1\ 1\ 0]$ and Fig. 4(b) with $H//[1\ 1\ 0]$, and related discussions in the paragraph before “In summary”. In particular, the level crossing observed in Fig. 4(f) near 1.5T was associated to the peak in Fig. 4(b) by the authors, however the magnetic fields are along different directions. Are the authors mixing the directions of the fields, or am I simply mis-reading the authors’ message?

Version 1:

Reviewer comments:

Reviewer #1

(Remarks to the Author)

The authors have replied part of questions mentioned in the initial review, the manuscripts are improved but they still did not reply the concerned questions:

1) Since the exotic magnetic behaviors of $\text{Yb}_2\text{Be}_2\text{SiO}_7$ is located at temperature below 0.5 K and field ($B < 1\text{T}$), the anisotropic magnetization and specific heat characterizations at $T < 0.5\text{K}$ and $B < 1\text{T}$ are necessary to check whether this exists field-induced phase transition, that can identify whether the system is a pure isolated spin dimer or interacted spin dimer. Even the neutron diffraction is measured but it is under zero field. The authors neglected this issue in their revision, I still proposed the authors to characterize the related properties.

As pointed by Reviewer 3, if this is a pure isolated spin dimer, the present results are trivial. But, the key is to investigate the magnetic behaviors at lower fields, while most of experimental results are missing.

2) The proposed dimer state is not convincing. In Fig. S7, one can not distinguish between model 2 and model 4. Moreover, both models cannot reproduce the specific data satisfactorily, especially in nonzero magnetic fields. The poor resolution, or intrinsic broadening of the spectra renders a separation of the 0.11 and 0.19 meV excitations nearly impossible. Therefore, a comparison of the Q dependence of these excitations is not convincing. The origin of the excitations at 0.7 and 1.2 meV remains elusive. Could that be observed with different EIS?

3) The Weiss temperature along the $[1\ 1\ 0]$ direction is very large compared to that along the $[0\ 0\ 1]$ direction. But the extracted exchange parameters show a larger J_z than J_y . How to reconcile these results?

Some minor points:

1) in Fig. S3, the 0.19 meV peak seems to locate above 0.2 meV by eye. Is the fit reliable?

2) Also, the peak width seems to be Q dependent, see Fig. 3. Is the large integration range in Fig. S3 justified?

Reviewer #3

(Remarks to the Author)

I have reviewed the authors' response along with the updated manuscript, and found the changes to be satisfactory. Therefore, I now fully support the publication of this paper without reservation.

Response to Referees

We thank the referees for their constructive and unique reports. We address their criticisms and outline the changes that we made to the manuscript in point form here:

Referee 1:

1) Yb₂Be₂SiO₇ possibly realizes the anisotropic spin dimer physics beyond isotropic Heisenberg model. But, the observed anisotropic magnetization behaviors can't be simply ascribed to the anisotropic exchange interactions since it can be related to the single ion effect of Yb³⁺ ions because similar anisotropy is observed at 2 K far above the energy scale of exchange interactions. Since broad peak centered at 0.5 K, the anisotropic magnetization characterization far below 0.5 K is necessary to check whether this is field-induced phase transition, that can identify whether the system is isolated spin dimer or interacted spin dimer. Without this characterization, the authors can't give the conclusion this is isolated system and exclude the $\sqrt{2}(|\uparrow\downarrow\rangle - |\downarrow\uparrow\rangle)$ ground states. Also, in the specific heat data, no experimental data is provided at the field regions between 0-1T, which can give some information on CEF level crossing for spin dimer. We proposed the authors to characterize the related properties.

There may be some confusion here between spin-1/2 and effective spin-1/2 quantum dimer magnets, which can have very different properties. To help alleviate this confusion, we have briefly commented on the differences between these two cases in the introduction of the revised manuscript and we have added a separate discussion section at the end of the paper where we go through this comparison more thoroughly.

In the pure spin-1/2 case, the dominant energy scale will be intradimer Heisenberg exchange which is usually antiferromagnetic. This isolated dimer Hamiltonian yields the familiar $(|\uparrow\downarrow\rangle - |\downarrow\uparrow\rangle)$ entangled ground state. Effective spin-1/2 moments are generated in materials with strong crystal field effects where the single ion ground state doublet is well-separated from the first excited level. For rare earth systems, the ground state doublet wavefunctions are a linear superposition of the different J_z basis states. These wavefunctions generate an anisotropic g-tensor, which leads to the anisotropic magnetization that persists above 2 K for Yb₂Be₂SiO₇. The effective spin-1/2 moments interact via an anisotropic exchange Hamiltonian rather than the more familiar Heisenberg Hamiltonian. The anisotropic intradimer exchange is usually the second-highest energy scale for effective spin-1/2 quantum dimer systems, so its effect on the magnetic properties is not apparent until lower temperatures. For example, the dimer level crossing observed in the magnetization data for Yb₂Be₂SiO₇ is only observed in the 0.4 K data when the temperature is cooled below the energy scale for the anisotropic intradimer exchange.

We investigated the origin of the broad peak centered at 0.5 K in the heat capacity data by measuring neutron powder diffraction patterns both above and below this temperature. The main results are presented in Fig. 1 (e) and (f) of the main manuscript. As we discuss in the text, there is no evidence for magnetic Bragg peaks in our data at 0.25 K and therefore we have ruled out the possibility of a phase transition to a ground state with long-range magnetic order. Neutron diffraction is an effective technique for ruling out the presence of low-temperature long-range magnetic order because it is a volume-integrated probe that is insensitive to small magnetic impurities. We also note that neutron spectroscopy is an effective technique for establishing an isolated (or weakly interacting) vs strongly interacting dimer system and low-T magnetic susceptibility data is not required for this purpose. Finally,

we did not collect additional specific heat data between 0 and 1 T because there is no evidence of anything exotic (e.g. a dimer level crossing) happening in this regime from both our 0.4 K magnetization data and our energy level diagrams presented in Fig. 4(b) and Figs. 4(e-g) respectively.

2) If the ground state is $|\psi_0\rangle = \frac{1}{\sqrt{2}}(|\uparrow\uparrow\rangle - |\downarrow\downarrow\rangle)$ singlet state, the magnetic susceptibility should drop down to zero value as temperature approaches zero Kelvin. How about the susceptibility data? The size of energy gap between the singlet state and first excited state?

This is not true. Instead, the magnetic susceptibility will level off at a finite value as $T \Rightarrow 0$ K regardless of the field direction. This behavior is related to the lack of a low-field magnetization plateau for this dimer ground state. The referee is correct that the magnetic susceptibility will go to zero as $T \Rightarrow 0$ K, regardless of the field direction, for the more familiar $(|\uparrow\downarrow\rangle - |\downarrow\uparrow\rangle)$ dimer ground state. The $(|\uparrow\downarrow\rangle + |\downarrow\uparrow\rangle)$ state makes an interesting comparison, since it exhibits hybrid behavior between these two cases. To ensure that these three possibilities are discussed in the manuscript, we have added panels to Fig. S6 that present the simulated low-T susceptibility data for three models with different dimer ground states. We have also revised Fig. 1(d) to plot χ vs T in the lowest-T regime, which includes new AC susceptibility data down to 50 mK, since this format makes it more obvious that the susceptibility data for $\text{Yb}_2\text{Be}_2\text{SiO}_7$ does not match qualitative expectations for a $(|\uparrow\downarrow\rangle - |\downarrow\uparrow\rangle)$ dimer ground state. The energy gap between the singlet state and the first excited state is 0.11 meV, which was measured directly by neutron spectroscopy.

3) For the Neutron scattering of $\text{Yb}_2\text{Be}_2\text{SiO}_7$, the powder samples are used that cannot give the dispersion relation of spin excitation, as presented in Figs. 2 and 3. The experimental data is enough to determine the CEF levels, the resolution is not high enough that the dispersionless nature of excitation can not be concluded based on the present data. And, how to understand the double mode at 0.11 meV and 0.19 meV, the related experimental measurements on single crystal is appreciated to clarify this experimental behavior. Only in that case, the excite states can be fully confirmed alongside their proposed model.

The referee is correct that powder samples cannot be used to measure dispersion relations of spin excitations. For quantum dimer systems measured by neutron spectroscopy, a common approach is to model the data with an isolated dimer Hamiltonian first. This is exactly what we did in this manuscript. The agreement between the powder-averaged data and the simulation with instrumental energy resolution included, presented in Fig. 4(a) and (b) respectively, is excellent. This implies that our isolated dimer Hamiltonian describes the physics of $\text{Yb}_2\text{Be}_2\text{SiO}_7$ well and additional terms (i.e. interdimer and/or dipolar interactions) will be small by comparison. We attribute the double-mode at 0.11 meV and 0.19 meV to two different single dimer excitations with degeneracies of 2 and 1 respectively. This eigenvalue scheme is further supported by our zero-field heat capacity simulations, as it explains the broad maximum in this data perfectly [see Fig. S5(b)]. The XYZ dimer model for effective spin-1/2 quantum dimer magnets can yield up to three magnetic excitations in neutron spectroscopy, unlike the more familiar Heisenberg model which will only yield one mode with a degeneracy of 3.

4) The spin dimer physics in Ce/Yb based SSL compounds included the proposed singlet state have been discussed in the similar compounds $\text{BaCe}_2\text{ZnS}_5$ [arXiv:2412.17913,2024] and $\text{Yb}_2\text{Be}_2\text{GeO}_7$ [Phys. Rev. B 110, 144445 (2024)]. The similar claim is reported, the related work should be cited.

We thank the referee for identifying these two important references. While we had already cited the $\text{BaCe}_2\text{ZnS}_5$ manuscript in our work, we missed the PRB paper on $\text{Yb}_2\text{Be}_2\text{GeO}_7$. We have now cited this paper in our manuscript and briefly discussed the results presented there.

5) In Figure S7 of supporting information, we don't observe the double broad peaks in the specific heat near critical field for closing the gap of spin dimer. While, the proposed theoretical model yield the two humps in $C_p(T)$ curves, this contradicts with the formation of isolated spin dimer, how to explain this controversy.

The referee is correct that our isolated dimer model does not describe the in-field heat capacity data near the 1.5 T level-crossing for H || ab-plane very well. This discrepancy arises from our failure to identify the appropriate sub-leading longer-range interaction term in our Hamiltonian, which could be due to interdimer exchange or a dipolar interaction. Since the neutron powder spectroscopy and the other bulk characterization data is described well by the isolated dimer model, we are confident that this longer-range interaction term will be a perturbation to our isolated dimer Hamiltonian. The best way to establish the form and magnitude of this interaction is to perform single crystal neutron spectroscopy on $\text{Yb}_2\text{Be}_2\text{SiO}_7$, which is beyond the scope of the current manuscript. Our inability to say more about this weak term does not affect the major conclusions of our paper, which are that $\text{Yb}_2\text{Be}_2\text{SiO}_7$ hosts a novel dimer ground state due to strong spin-orbit coupling. In other words, the addition of a sub-leading term to our magnetic Hamiltonian will not alter the zero-field magnetic ground state of this system, which is the most important finding in our paper. A new, separate discussion section near the end of our manuscript provides more information on how this exotic, entangled state-of-matter is stabilized and how its identification in this system has broad implications for other rare-earth-based quantum dimer magnets, including Shastry-Sutherland systems, which are currently a subject of great interest in the condensed matter physics community.

Referee 2:

We find that the quality of the presentation of these results has room for improvement in order to be understandable by a broad audience not familiar with the plethora of neutron scattering techniques used in the paper. In addition, the potential impact that these new results have, could be motivated further, especially given the wealth of literature on SSM physics and other recent experimental efforts on crystal systems featuring SSM physics. For these general reasons we suggest that the authors rework the paper before we can support it for publication.

We thank the referee for his/her constructive review and the valuable suggestions on how to improve the quality of the manuscript. For the first comment, we are assuming that the referee would like us to implement the recommended changes below so that's what we have done. We have also added some discussion to the manuscript that describes the difference between true spin-1/2 vs effective spin-1/2 quantum dimer systems since we've realized now that this would help a broad audience understand the significance of our results more easily. As for the second comment, it is an excellent point. We have added some discussion near the end of the manuscript that explains why our results are important and relevant for the plethora of recent work on rare-earth-based Shastry Sutherland magnets.

On the results:

-Abstract: Revise the sentence "a singlet ground state with the symmetric wavefunction"

We have modified this sentence, so it now presents the wavefunction in the less ambiguous form $(|\uparrow\downarrow\rangle - |\downarrow\uparrow\rangle)$.

-Introduction and conclusions: link results to existing theoretical work

We have added some theory references to the introduction which discuss Hamiltonians (i.e. Ising and XXZ models) relevant for anisotropic SSL systems. We have also added a separate discussion section to the end of the paper that explains the difference between spin-1/2 and effective spin-1/2 quantum dimer magnets, provides more details of the XYZ dimer theory relevant to our material, and explains how our results are relevant for all effective spin-1/2 quantum dimer magnets including many new rare-earth-based SSL systems.

Page 5: It is not stated in the text how the inelastic neutron scattering data (proportional to $S(Q,E)$) is compared to equations (2) and (3). Is the partially Q -integrated powder data integrated along E in order to yield the total scattering intensity $S(Q)$? In Ref. [48] the term “exclusive structure factor” is used to refer to the Q -dependence of $S(Q,E)$ at a fixed E corresponding to a selected transition from the ground state. Is this definition adopted in the paper? If so it should be made explicit.

This is a good point. We have rewritten this section of the manuscript to clarify how we compared the two different dimer structure factors to the most intense mode of the neutron spectroscopy data. We took a constant- E cut of the $E_i = 1.55$ meV data centered at 0.11 meV with an integration range of ± 0.05 meV and found that the Q -dependence fit much better to the expected structure factor for the six $S_z \neq 0$ solutions. This is compelling evidence that $\text{Yb}_2\text{Be}_2\text{SiO}_7$ does not have a $(|\uparrow\downarrow\rangle - |\downarrow\uparrow\rangle)$ dimer ground state. We are not familiar with the term “exclusive structure factor” and it is not commonly used by the neutron scattering community, so we have chosen not to adopt it here.

Equation (2) is derived in Ref. [48]. The authors do not derive explicitly nor provide a reference that leads to a Q -dependence of the form shown in Eq. (3) (Eq. (6) in the SM). Is this equation rather a phenomenological structure factor that goes to a constant value as $Q \rightarrow 0$ as seen in experiment?

This is another good point. Upon review of our discussion related to those two equations, we agree with the referee that it was misleading and unnecessarily vague. We have rewritten this section of the manuscript to improve clarity. For the XYZ dimer model, the Q -dependence of the most intense excitation is not described by an analytical function over a broad Q -range. However, for the 12 XYZ dimer models being considered in this work the Q -dependence of this excitation is well-described by either Eq. (2) or Eq. (3) in the low- Q regime ($\leq 1.6 \text{ \AA}^{-1}$). Examining the Q -dependence of the most intense mode in the neutron spectroscopy carefully allows us to reduce the number of viable solutions from 12 to 6. When working to address this point, we found an error in our code for powder-averaging the dimer structure factors. We have now corrected this error and re-generated Fig. 3, Fig. S6, and Fig. S7. We have validated the accuracy of our new powder-averaged results by using the community software Sunny (<https://github.com/SunnySuite/Sunny.jl>) to reproduce the calculations with our own code. After correcting this powder-averaging error, we are now convinced that we can no longer use our neutron powder spectroscopy data to determine the magnetic Hamiltonian of $\text{Yb}_2\text{Be}_2\text{SiO}_7$ unambiguously. There are two different models that show similar agreement between the simulations and our bulk characterization and neutron powder spectroscopy data. Nonetheless, they both have $S_z \neq 0$ dimer ground states so our major finding that $\text{Yb}_2\text{Be}_2\text{SiO}_7$ hosts a magnetic ground state with novel bipartite entanglement is unaffected. We have modified the manuscript in the appropriate places to ensure that our modified conclusions are properly communicated.

Is there a physical reason why the intradimer xx -exchange is ferromagnetic whereas the zz -exchange is antiferromagnetic? Could the sign of the exchange tensor elements be determined from the data that entered the fit? In other words, did the fit procedure allow to determine whether the exchange is AF or AFM?

This is a good question. Phrased more generally, one could ask if there are simple physical explanations for the relative magnitudes and signs of each component in an anisotropic exchange matrix. We are not aware of straightforward connections between the crystal structure, single ion properties, and an anisotropic exchange matrix along the lines of Goodenough-Kanamori rules for isotropic exchange in transition metal magnets. The second question is easier to answer. There are only 12 anisotropic intradimer exchange matrices consistent with the eigenvalue scheme determined from neutron spectroscopy and zero-field heat capacity data. In other words, the three components of these matrices satisfy Supplementary Eq. S2. Each solution has two larger components with equal magnitudes and a smaller component. The signs of the different components are variable. Therefore, our procedure for identifying the best solution from these 12 possibilities does indeed allow us to determine if each component (J_{xx} , J_{yy} , and J_{zz}) is antiferromagnetic or ferromagnetic. In fact, the sign of these terms helps to establish the single dimer ground state of the system.

A discussion on the results put in the context of Refs [39, 49 and 51] would be beneficial. Especially given that the ground state wavefunction of $\text{Yb}_2\text{Be}_2\text{SiO}_7$ deduced from the heat capacity and INS data appears identical to the one found by the authors of Ref. [39] for their model system.

This is a great suggestion. We have now added a discussion section with this content near the end of the manuscript.

Page 4: specify quantitatively how good the agreement is between the measured polycrystalline magnetization and the calculated powder-averaged magnetization.

The refined values from the local site susceptibility measurement are nearly the same as the values from our powder-averaged magnetization measurements. We have now compared the two sets of values directly in the manuscript.

On the presentation:

Discussion regarding the g-tensor in page 2 can be left for after the pNDP results and discussion.

This is another good point. We have moved the g-tensor discussion to the single ion properties section of the paper to help explain why the local site susceptibility measurement was useful for this system.

Fig.2(f) clarify meaning of values (0.28,0.52,1.03) in figure caption.

These are the measured values from the powder-averaged magnetization. We have now stated this explicitly in the figure caption.

As measurements were made both in powder and single crystal samples, specify the sample form for the results in the main text, eg. "Powder neutron spectroscopy" instead of "Neutron spectroscopy"

We have now specified if each measurement was made on powders or single crystals the first time it is mentioned in the manuscript. We have also provided these details in the methods section.

Specify instrumental energy resolution for/or in Fig.2(c)

We now explicitly state in the manuscript text that the $\text{Er}_2\text{Be}_2\text{SiO}_7$ crystal field levels shown in Fig. 2(c) are nearly resolution limited. We also revised Fig. 2(c) to plot the correct constant-Q cut for $\text{Er}_2\text{Be}_2\text{SiO}_7$ to facilitate a direct comparison to the $\text{Yb}_2\text{Be}_2\text{SiO}_7$ dataset collected under the same experimental conditions. Finally, we provided instrumental energy resolution information used to generate the

simulation shown in Fig. 3(b). While the referee did not explicitly ask for this last change, we realize that it was an oversight not to provide this information previously.

For the published version, the paper would benefit greatly from higher-quality (higher-DPI) figures throughout the text. The xyz coordinate axes in Fig. 1 for example are barely legible in a printed version of the paper.

Yes, this is a good point. We have re-generated all the figures with a DPI of 300.

Supplement, section II. The equation relating the local susceptibility tensor to the magnetization and applied field should be written in component form to avoid a notation that implies division by a vector.

Supplement, figure S6. The x-label for panels (a), (c) and (e) should read $Q (\text{\AA}^{-1})$.

Thanks for pointing out these two errors. They have been corrected in the revised manuscript.

Referee 3:

However, I am not in a positive position in supporting the publication of the manuscript in Nature Communications, at least not in the current form. In fact, in contradiction to what the authors have claimed in the title, the results of this manuscript show that this compound has “boring disentangled” dimer state, and despite the lattice structure the physics of this material has almost NOTHING to do with “Shastry-Sutherland” physics. As the authors have successfully demonstrated, the physics of this material is a simple “molecular”-like paramagnet, which can be described by isolated (disentangled) dimers (molecules).

I believe that both the authors and the editor would agree with me that reporting negative results is also a helpful way in maintaining the healthy growth of the scientific community. I understand that the authors were trying to find a fancy title so the manuscript could have a higher chance in glossy journals like the Nature series. However, in this particular case, it seems to me it does not hurt to publish this paper with an unbiased title in Nature Communications. Due to the increasing interest in finding novel Shastry-Sutherland physics in rare-earth based systems, this paper could save the global research community working on related compounds a lot of money/time, by simply letting people know that there is no Shastry-Sutherland physics in this particular compound (and likely also in sibling compounds). In this sense, publishing this paper with an unbiased message in Nature Communications will be greatly helpful to the community. My suggestion is to modify the title and the discussions of the paper to unbiasedly reflect the physics of this material, although it is in some sense a “negative” result.

We sincerely thank the referee for his/her overall positive review of our work. While the referee seems to understand our general rationale for trying to publish this work in a high-profile journal like Nature Communications, our choice of some unfortunate language has led to his/her main criticism. We realize now that the term ‘entangled dimer state’ is misleading as it could imply that there is entanglement between dimers, which would be a more exotic state-of-matter than the one we have found for $\text{Yb}_2\text{Be}_2\text{SiO}_7$ here. It was not our intention to convey this message in our paper as it is not supported by our experimental data or our simulations. To improve clarity, we have modified our title to make it clear that we have intradimer (or ‘bipartite’) entanglement in our system only. We think that this entangled state-of-matter is still quite novel as it has only been observed in one other quantum dimer magnet quite recently (the manuscript is still unpublished) and many condensed matter physicists still don’t seem to appreciate that it can be stabilized in effective spin-1/2 quantum dimer magnets by anisotropic intradimer exchange. Instead, it is often assumed that all quantum dimer magnets must have $(|\uparrow\downarrow\rangle -$

$|\downarrow\uparrow\rangle$) dimer ground states due to the plethora of true spin-1/2 systems with antiferromagnetic Heisenberg exchange that have been characterized.

Regarding the referee's related comment about Shastry-Sutherland physics, we agree that the applicability of the J_1 - J_2 SSL Hamiltonian to $\text{Yb}_2\text{Be}_2\text{SiO}_7$ is an open question since our work shows that this system is very close to the isolated dimer limit. Since the simple isolated dimer model does not describe all our data perfectly, we know that we are missing a sub-leading term in the Hamiltonian that may arise from interdimer exchange (i.e. J_2 in the SSL model) or a dipolar interaction. However, it is beyond the scope of our work to conclusively determine the nature of this term, and it is not relevant for our main conclusions that the referee stated in his/her review. To address the referee's concern, we have decided not to explicitly call $\text{Yb}_2\text{Be}_2\text{SiO}_7$ a Shastry-Sutherland magnet in the manuscript and instead we refer to it as a 'quantum dimer magnet' now. Our understanding is that a quantum dimer magnet is any system with structural dimers where the pairs of spins entangle to form a singlet state, which is certainly true in the present case. This modification also allowed us to discuss the difference between pure spin-1/2 and effective spin-1/2 quantum dimer magnets in the introduction and a new discussion section of our manuscript and explain the significance of our results more clearly in the same discussion section.

Another minor note: I am confused by comparing Fig. 4(f) with $H//[-1\ 1\ 0]$ and Fig. 4(b) with $H//[1\ 1\ 0]$, and related discussions in the paragraph before "In summary". In particular, the level crossing observed in Fig. 4(f) near 1.5T was associated to the peak in Fig. 4(b) by the authors, however the magnetic fields are along different directions. Are the authors mixing the directions of the fields, or am I simply misreading the authors' message?

Thanks for catching this issue. There are two dimer sublattices with the same $H \parallel [001]$ energy level diagrams and opposite behavior along the $H \parallel [110]$ and $H \parallel [-110]$ directions. We have now plotted the energy level diagrams for the other sublattice since the level crossing then occurs when $H \parallel [110]$, which agrees with the field direction convention we chose for plotting the bulk characterization data. We have also tried to clarify the sublattice-dependent level crossing behavior in the manuscript text.

Extra correction:

One additional correction we made not related to any referee comment was to revise Fig. S5. We accidentally included a variable scale factor in the heat capacity simulations presented in the previous version of this figure, which doesn't make physical sense since the simulation result should be in absolute units. Now that this error has been corrected, most of the models we previously discarded using Fig. S5 have even larger discrepancies with the experimental data. Resolving this issue does not affect any of our conclusions.

Thanks again to the referees for their careful reviews of our work, which helped to significantly strengthen the revised manuscript.

Sincerely,

Dr. Adam Aczel, Corresponding Author

Response to Referees

We thank the two referees for their second round of reports. We address Referee 1's remaining criticisms and outline the changes that we made to the manuscript in point form here:

Referee 1:

1) *Since the exotic magnetic behaviors of Yb₂Be₂SiO₇ is located at temperature below 0.5 K and field ($B < 1T$), the anisotropic magnetization and specific heat characterizations at $T < 0.5$ K and $B < 1T$ are necessary to check whether this exists field-induced phase transition, that can identify whether the system is a pure isolated spin dimer or interacted spin dimer. Even the neutron diffraction is measured but it is under zero field. The authors neglected this issue in their revision, I still proposed the authors to characterize the related properties. As pointed by Reviewer 3, if this is a pure isolated spin dimer, the present results are trivial. But, the key is to investigate the magnetic behaviors at lower fields, while most of experimental results are missing.*

During the first round of revisions, we already added anisotropic magnetization data plotted as $\chi = M/H$ below 0.5 K (see revised Fig. 1d). In the initial manuscript, we also included 0 T heat capacity data down to dilution fridge temperatures. Both datasets find no evidence for long-range magnetic order in zero field and their T -dependence is consistent with the bipartite-entangled dimer ground state that we have proposed in this work. We also have dilution fridge heat capacity data in selected magnetic fields (1 T, 2 T, 3 T) for various sample orientations that show no evidence for field-induced magnetic order. We have added one sentence to the second revision of the manuscript on p. 6-7 to make this clear, which states: "It is worth noting that there is no evidence for field-induced magnetic order in the heat capacity measurements down to dilution fridge temperatures."

We assume that the referee is focused on the search for field-induced order between 0 and 1 T due to the small discrepancy between the $H \parallel [110]$ heat capacity data and the isolated dimer simulation for 1 and 2 T. As we have already pointed out in our first referee response letter and the first revision of the manuscript on p.7, we attribute these discrepancies to the proximity of that heat capacity data to the 1.5 T level crossing identified in both our magnetization data and the isolated dimer modeling. This is precisely the regime where one expects the heat capacity of an isolated dimer model and a weakly-interacting dimer model to have the largest discrepancies, as the spin gap between the two lowest dimer levels becomes comparable to the weak interdimer interaction. The nearly perfect agreement between the 0 T and 3 T $H \parallel [110]$ heat capacity data and simulations, where we are far away from the level crossing, validates this point. Note that the differences between the H - T phase diagrams of the isolated and weakly-interacting dimer models have been investigated extensively in pure spin-1/2 quantum dimer magnets. For good review articles, please see Giamarchi et al, Nature Physics 4, 199 (2008) and Zapf et al, Rev. Mod. Phys. 86, 563 (2014). The relevant text in the manuscript states: "It is worth noting that the largest discrepancy in the $H \parallel [110]$ bulk characterization data and the simulations is observed in the fixed-field heat capacity measurements shown in Fig. 4(d) near the expected critical field for the level crossing. This is likely due to neglecting the appropriate sub-leading interaction in the simulations, which should have the largest effect on the magnetic properties of the system in this regime. Indeed, neutron powder spectroscopy also suggests that a small additional term should be added to the Hamiltonian to capture the increased bandwidth of the 0.11 meV mode observed in the data, but its accurate determination without single-crystal neutron spectroscopy data is challenging. The nature of this term is not obvious either, as the expected sub-1 K energy scale means it

could arise from interdimer exchange or a dipolar interaction. While additional theoretical work on candidate magnetic Hamiltonians for this system and its isostructural family members could help to address this issue, neglecting this extra term does not affect the novel zero-field magnetic ground state for Yb₂Be₂SiO₇ established here.”

The small deviations between the $H \parallel [110]$ heat capacity data and simulations at 1 and 2 T show that Yb₂Be₂SiO₇ is likely a weakly-interacting dimer system, which we already acknowledged in the first revision of the manuscript in several places. We are hesitant to include an interdimer term in our Hamiltonian for the simulations because it is very weak, its origin is unclear from our data (i.e. in-plane exchange, out-of-plane exchange, dipolar interaction, or some combination of these terms) and it would not be possible to solve the revised Hamiltonian exactly, so we would have to resort to more complicated numerical approaches for the simulations that wouldn't add significant value to the paper and instead likely reduce the credibility of our analysis instead. **Importantly, the main conclusion of our work is that Yb₂Be₂SiO₇ has an exotic, entangled dimer ground state in zero field. This is true for both isolated dimer and weakly-interacting dimer scenarios. Identifying field-induced magnetic order in a small region of the H-T phase diagram (most likely between 1 and 2 T for $H \parallel [110]$) doesn't affect this conclusion and we don't consider it to be a very impactful result.** We also don't expect to observe field-induced order for any applied field. Beyond the lack of experimental evidence for this field-induced order via our heat capacity measurements in selected applied magnetic fields, the out-of-plane Curie-Weiss temperature is only -0.1 K, which implies that the energy scale for the out-of-plane interdimer exchange term necessary for this 3D field-induced order is barely above the lowest experimentally accessible temperature provided by a dilution fridge. This suggests that Yb₂Be₂SiO₇ is a weakly-interacting dimer system with the largest interdimer term likely arising from in-plane exchange as expected for the 2D Shastry-Sutherland model. Note that not all weakly-interacting dimer models generate magnetic order somewhere in their H-T phase diagrams.

As we already described in the first revision of the manuscript, particularly in the discussion section, the zero-field dimer ground state of Yb₂Be₂SiO₇ is far from trivial. In fact, we think that this entangled state-of-matter is quite novel and precisely what makes this work worthy of publication in Nature Communications. It has only been observed/proposed in one other quantum dimer magnet quite recently (the manuscript is still unpublished) and many condensed matter physicists still don't seem to appreciate that it can be stabilized in effective spin-1/2 quantum dimer magnets by anisotropic intradimer exchange. Instead, it is often assumed that all quantum dimer magnets must have $(|\uparrow\downarrow\rangle - |\downarrow\uparrow\rangle)$ dimer ground states due to the plethora of true spin-1/2 systems with antiferromagnetic Heisenberg exchange that have been characterized. This confusion has led to significant challenges with characterizing new dimer systems with strong spin-orbit-coupling, which have come to the forefront as a topical research area in the field. In fact, some researchers are proposing materials in this family as new quantum spin liquid candidates that may not order for similar reasons to Yb₂Be₂SiO₇ instead.

2) *The proposed dimer state is not convincing. In Fig. S7, one cannot distinguish between model 2 and model 4. Moreover, both models cannot reproduce the specific data satisfactorily, especially in nonzero magnetic fields. The poor resolution, or intrinsic broadening of the spectra renders a separation of the 0.11 and 0.19 meV excitations nearly impossible. Therefore, a comparison of the Q dependence of these excitations is not convincing. The origin of the excitations at 0.7 and 1.2 meV remains elusive. Could that be observed with different Eis?*

We'll assume that the referee doesn't think the proposed dimer state is convincing for the specific reasons given here and address each of these points in turn.

First point - In the first revision of the manuscript on p. 6, we had already acknowledged that the data presented in our paper cannot be used to distinguish between models 2 and 4. As shown in Fig. S7, the simulated bulk characterization data (magnetization and specific heat) looks the same, as expected. The strongest mode in neutron spectroscopy is also the same for the two models. It turns out that the only way to differentiate between these two models is via the Q-dependence of the two weak modes in neutron spectroscopy, but one weak mode is at the same energy transfer as the strong mode (0.11 meV) and the second weak mode (0.19 meV) is not sufficiently well-separated for us to draw any conclusions from it. Despite both models providing equivalent descriptions of our data, our main conclusion shown above in the first bold sentence is still robust. The only drawback now is that we are not certain if the dimer ground state of the system is $|\uparrow\uparrow\rangle - |\downarrow\downarrow\rangle$ or $|\uparrow\uparrow\rangle + |\downarrow\downarrow\rangle$. We don't think that this diminishes the impact of our work. The relevant manuscript text states: "Two of the solutions show superior agreement with the bulk characterization data. Although the Q-dependence of the weak mode intensities from neutron powder spectroscopy is not exactly the same for these two models, our data is not sufficient for differentiating between them conclusively."

Second point - The two models explain the 0 T heat capacity data and the H || [110] heat capacity data at 3 T very well. There are two separate issues leading to small discrepancies between the intermediate field H || [110] heat capacity data and the models and the H || [001] heat capacity data and the models. Both issues were already discussed in the first revision of the manuscript on p. 7. The reason for the first discrepancy along the H || [110] direction is also provided in our response to (1) above. The reason for the second discrepancy along the H || [001] direction is that there was a modest misorientation of the sample for that measurement. To verify this, we had already collected additional H || [001] heat capacity data over a limited temperature range at 2 T and 3 T. The associated text on p. 7 of the revised manuscript read as follows: "The agreement between experiment and theory is excellent, with the small differences between the H || [001] heat capacity data and the simulation arising from modest sample misalignment that was confirmed by re-measuring the 2 T and 3 T data with coarse temperature steps for $T \geq 1.8$ K." To emphasize this new data further, we have now added the following to the end of this sentence: "...which are shown as points connected by dashed curves in Fig. 4(c)." We have also referred to this new heat capacity data explicitly in the figure caption.

Third point – We agree with the referee that the neutron spectroscopy experiment does not have sufficient energy resolution to separate the modes at 0.11 meV and 0.19 meV to examine their Q-dependences independently. The critical point here is that the simulated intensity of one mode is significantly larger than the intensity of the other two modes for all 12 isolated dimer models with the appropriate eigenvalues (see Fig. S6 for three examples), so we can compare the simulated Q-dependence of this intense mode to the experimental data quite accurately. We use both the measured 0.11 meV eigenvalue and the Q-dependence of the intense mode to eliminate 8 of the 12 isolated dimer models from consideration. We have added two sentences to the manuscript to explain how the Q-dependence of the intense mode can be compared between experiment and theory, which read as follows: "Importantly, all 12 models have one intense mode and two much weaker modes. This common feature allows us to reliably compare the Q-dependence of the intense mode for the models with the neutron spectroscopy data even though neither weak mode is completely resolved from the intense one."

Fourth point – We agree with the referee that we couldn't conclusively identify the origin of the weak 0.7 and 1.2 meV magnetic excitations. They are not spurious, as the 0.7 meV mode was observed in both the $E_i = 1.55$ meV and 2.49 meV data shown in Fig. S4 and Fig. 3(c) respectively. We added one sentence to the second revision of the manuscript to make this more explicit now: "We first verified that none of these modes were spurious by measuring them with different incident energies." We already discussed the possible origin of these modes at length in the first revision of the manuscript on p. 5. The manuscript

text reads as follows: “Since effective spin-1/2 dimer models with strong intradimer exchange anisotropy can have a maximum of three single-dimer excitations, it is clear that all four modes (at 0.11, 0.19, 0.7 and 1.2 meV) observed here do not have the same origin. Zero-field heat capacity only depends on the eigenvalues of these models, so we compared our data shown in Fig. 2(a) to simulations assuming various energy-level schemes (see Supplementary Fig. S5). The best simulation of the zero-field heat capacity data corresponds to an isolated dimer model with a doubly-degenerate mode at 0.11 meV and a non-degenerate mode at 0.19 meV. This implies that the two higher-energy modes have a different origin. In fact, as shown in Fig. 3(d) their Q-dependence is well-described by the Yb^{3+} magnetic form factor squared multiplied by the known structure factor of a singlet-triplet transition for an isolated Heisenberg dimer model with the square plaquette distance $d_{\text{nnn}} = 3.84 \text{ \AA}$. Since these higher-energy modes have a negligible contribution to the specific heat, this implies that only a small fraction of Yb^{3+} ions form longer-range dimers in $\text{Yb}_2\text{Be}_2\text{SiO}_7$, possibly due to the known Be/Si site mixing in this system. We also compared the Q-dependence of these modes to expectations for the lowest-energy excitations of rectangular tetramers, but this model could not account for the Q-position of their maximum intensities. Finally, we note that higher-energy excitations in $\text{SrCu}_2(\text{BO}_3)_2$ have been attributed to singlet bound states consisting of contributions from multiple dimer units and a similar alternative origin for the higher-energy modes cannot be ruled out in the present case.”

3) The Weiss temperature along the [110] direction is very large compared to that along the [001] direction. But the extract exchange parameters show a larger J_z than J_y . How to reconcile these results?

There may be some confusion here between different types of anisotropic exchange interactions. One common type of exchange anisotropy arises in 2D magnets, where the in-plane exchange interactions are stronger than the out-of-plane interactions. Another type of exchange anisotropy occurs between a single pair of magnetic ions due to strong spin-orbit-coupling, where the exchange interaction needs to be described by a tensor rather than a single value that is appropriate for the Heisenberg case. Both types of anisotropic exchange play a role in $\text{Yb}_2\text{Be}_2\text{SiO}_7$. Since the Yb ions form a Shastry-Sutherland lattice, the system is expected to be quasi-2D in nature. The measured Curie-Weiss temperatures are consistent with this scenario since the in-plane values are much larger than the out-of-plane value. The anisotropic intradimer exchange, given by the parameters $J_{\{xx\}}$, $J_{\{yy\}}$, and $J_{\{zz\}}$, is the driving force for the exotic entangled dimer ground state of the system. To help alleviate some of this confusion, we have added the following text to the manuscript on p. 3: “...and the larger in-plane Curie-Weiss temperatures are consistent with the expected quasi-2D behavior of $\text{Yb}_2\text{Be}_2\text{SiO}_7$. The small negative Curie-Weiss temperatures are indicative of weak antiferromagnetic interactions in the system under the assumption of isotropic exchange, which is not valid for this system as shown below.”

4) in Fig. S3, the 0.19 meV peak seems to locate above 0.2 meV by eye. Is the fit reliable?

The 0.19 meV peak position is deceiving by eye because this feature shows up as a shoulder in the data, which leads to a sizable error in the peak center value (see Table S4). We are confident that the fit is reliable though because the zero-field heat capacity data is described almost perfectly by a single dimer model with a doubly-degenerate excitation at 0.11 meV and a single excitation at 0.19 meV, as illustrated in Fig. S5b. If the 0.19 meV shoulder is not included in the heat capacity simulation, the agreement with the data is reduced significantly as shown in Fig. S5a.

5) Also, the peak width seems to be Q dependent, see Fig. 3. Is the large integration range in Fig. S3 justified?

There is essentially no dispersion in the intense, low-energy magnetic excitation, as expected for an isolated dimer or very weakly-interacting dimer model. This ensures that our constant Q-cut with a

relatively large Q integration range can be used to accurately determine the 0.11 and 0.19 meV peak positions. The excellent agreement between the zero-field heat capacity data and the simulation with this dimer energy scheme further validates our approach.

Thanks again to Referee 1 for his/her review of our work.

Sincerely,

Dr. Adam Aczel, Corresponding Author